# Same Graph Cross-Task Transfer in GNNs: Protocols and Predictors

**Neelam Akula** [* 1] **Surbhi Kumar** [* 1] **Murat Kantarcioglu** [2] **Baris Coskunuzer** [1]

## Abstract

Many real-world graphs support multiple predictive tasks over the same underlying structure, creating an opportunity to reuse supervision across node classification (NC) and link prediction (LP). However, existing evaluations often rely on incompatible splits, observed-graph assumptions, and negative sampling rules, making conclusions about same-graph cross-task transfer unreliable. We formalize same-graph NC–LP transfer and propose a leakage-free protocol that fixes node and edge splits, uses a shared message-passing graph that excludes evaluated edges, and employs fixed negatives for LP. Across three backbones (GCN, GraphSAGE, GPS), we find transfer is strongly directional and predictable: NC→LP is consistently beneficial on homophilic graphs, while LP→NC is fragile and can even degrade accuracy under naive representation reuse. LP→NC becomes reliably positive mainly in a structure-dominant regime where LP is easy but NC is unsaturated, suggesting LP acts as structural pretraining. Finally, we introduce CoTask Score (CTS) to summarize joint NC+LP utility when a shared encoder must serve both tasks, and show that simple dataset statistics, especially homophily, can guide mechanism choice and help avoid negative transfer.

## 1. Introduction

Graph representation learning is commonly developed and benchmarked around a single supervised objective, most often node classification (NC) (Kipf & Welling, 2017; Hamilton et al., 2017; Veličković et al., 2018) or link prediction (LP) (Zhang & Chen, 2018; Kipf & Welling, 2016). This single-task view has accelerated progress, but it mismatches how graphs are deployed in practice. Real systems answer multiple questions over the same underlying graph: they classify entities, predict missing or future links, rank candidates, and detect anomalies (Ying et al., 2018; Xia et al., 2021). For example, in social network graphs, a node classification task can be used to determine whether a given node represents a human user or a bot. In the same social network graph, link prediction can be employed to suggest potential friendship connections. Crucially, the required supervision signals often *coexist* on the same dataset: node labels for NC and observed edges for LP.

This creates two practical needs. First, since both supervision sources are available, can supervision from one task improve the other on the same graph? Second, many deployments prefer a shared encoder for NC and LP, to amortize training and serving costs and maintain a single embedding space, rather than optimizing each objective separately (Zhang & Yang, 2021; Ke et al., 2021). Yet standard pipelines train NC without using edge supervision beyond message passing and train LP without using node labels, leaving cross-task gains on the table.

Same graph cross-task learning remains poorly understood, largely because the evaluation is easy to get wrong. Standard NC and LP pipelines often use different split constructions, different observed graphs, and different negative edge sampling rules. These inconsistencies can create apparent gains that are actually protocol artifacts, or can hide real transfer effects behind confounding changes in the training signal. While large-scale benchmarks have standardized evaluation within individual task categories (Hu et al., 2020a), they do not address the multi-objective setting where node labels and link supervision are both available and intentionally leveraged across objectives on the same graph.

Beyond protocol issues, there are also structural reasons to expect transfer to be nontrivial. NC supervision encourages representations that separate label consistent neighborhoods, while LP supervision emphasizes pairwise compatibility and edge formation patterns. These signals can align, but they need not, suggesting that cross-task reuse is likely *directional* and *regime dependent*, and that successful reuse should be predictable from simple graph diagnostics (Wang et al., 2025; Platonov et al., 2023).

---

[*]Equal contribution [1]Department of Mathematical Sciences, University of Texas at Dallas, Richardson, TX, USA [2]Department of Computer Science, Virginia Polytechnic Institute and State University, Blacksburg, VA, USA. Correspondence to: Neelam Akula <neelam.akula@utdallas.edu>.

*Proceedings of the $43^{rd}$ International Conference on Machine Learning*, Seoul, South Korea. PMLR 306, 2026. Copyright 2026 by the author(s).

In this paper, we formalize **same graph cross-task transfer** between NC and LP and study it under a leakage-free, and standardized evaluation protocol. We fix node and edge splits, construct a consistent observed graph that prevents evaluation edges from appearing in message passing neighborhoods, and use fixed negatives (i.e., node pairs that do not appear as edges in the observed graph) for LP so that comparisons reflect genuine transfer rather than changes in sampling. Within this setting, we evaluate bidirectional transfer (NC→LP and LP→NC) across three representative backbone architectures (GCN, GraphSAGE, and GPS) under both transductive and inductive regimes. We further summarize *multi-objective* performance in scenarios where a single training regime must support both tasks.

Our experiments reveal a clear and practically important pattern: *cross-task transfer is strongly asymmetric, and the asymmetry is predictable.* Across backbones and settings, *NC→LP reliably improves LP on homophilic graphs*, where label similarity and connectivity are aligned. In contrast, *LP→NC is far less reliable*, and naive representation reuse can even degrade node classification. We also observe a second regime where LP→NC becomes beneficial: a *structure-dominant* regime, where edge prediction is already highly learnable from graph structure, while node classification remains unsaturated (high LP learnability with substantial NC headroom). In this setting, LP serves as structural pretraining for NC, improving accuracy even when homophily is low. These regimes can be anticipated from simple dataset statistics (e.g., homophily and baseline task learnability), making cross-task reuse a more predictable design choice.

In this work, rather than proposing a new architecture, our goal is to establish a clean evaluation paradigm for the same graph transfer learning tasks and to extract actionable guidance on when and which direction of reuse is worth attempting, and when a coupled regime yields a strong *single model* solution for both NC and LP.

**Our contributions.**

- **Problem setting: same graph cross-task learning.** We formalize NC and LP reuse on a single underlying graph with coexisting supervision signals, and we study both directions (NC→LP and LP→NC).
- **Leakage free evaluation protocol.** We introduce a standardized setup that fixes splits, observed graph construction, and LP negatives across all methods, enabling fair, reproducible comparisons and preventing neighborhood leakage from evaluation edges.
- **Systematic study across backbones and transfer strategies.** We evaluate five lightweight cross-task transfer strategies (WS, ET-Rep, ET-Concat, MV, Joint; defined in Section 4) across three representative backbones (GCN, GraphSAGE, GPS), and document a robust directional asymmetry that persists across architectures and regimes.

- **Multi-objective model selection and predictors.** We introduce CoTask Score (CTS) to summarize joint NC+LP utility when one regime must serve both tasks, and we relate transfer success to interpretable diagnostics: homophily predicts NC→LP gains, while LP→NC gains concentrate in structure dominant regimes with high LP learnability and NC headroom.

## 2. Related Work

**Multi-task learning and transfer on graphs.** Multi-task learning (MTL) shares statistical strength across related prediction problems through shared representations and task-specific heads, but can suffer from *negative transfer*, where optimizing for one task degrades performance on another under joint training or representation reuse (Caruana, 1997).

Recent work frames MTL as multi-objective optimization and mitigates gradient interference, for example via gradient projection (PCGrad) (Yu et al., 2020) or adaptive loss balancing (GradNorm) (Chen et al., 2018). In graph representation learning, transfer is often studied via auxiliary objectives or pretraining followed by task-specific finetuning, including mutual-information maximization (Veličković et al., 2019), contrastive learning (Zhu et al., 2020; You et al., 2020), masked reconstruction (Hou et al., 2022), and broader pretraining suites (Hu et al., 2020b). Other works couple node and edge signals with a shared encoder, for example via generative or reconstruction objectives that model adjacency alongside node attributes (e.g., VGAE (Kipf & Welling, 2016) and GPT-GNN (Hu et al., 2020c)). While related in spirit, these approaches are typically developed and evaluated for a particular objective (e.g., generative link reconstruction or masked prediction) and do not isolate *bidirectional* NC↔LP transfer on the *same* graph under a unified evaluation protocol. Moreover, when NC and LP are both reported, they often follow different conventions across tasks, including split construction, the representation-learning graph used for message passing, and negative sampling, which can confound cross-task conclusions.

We note that an important complementary direction is cross-graph and cross-domain generalization, where a model trained on one or more source graphs is evaluated on unseen target graphs (Wang et al., 2025). Our setting is orthogonal: we study same-graph transfer where both supervision signals coexist on a single fixed graph, and this case must be understood under controlled conditions before asking whether the patterns persist under distribution shift across graphs. Similarly, while we evaluate GCN, GraphSAGE, and GPS as representative backbones spanning message-passing and attention-based families, newer sequence-inspired or state-space graph architectures are an important frontier; we do not claim universality beyond the families studied here, and extending this analysis to such architectures is a natural

direction for future work. Finally, our protocol and findings are specific to the NC↔LP pair on a single graph; other task combinations such as graph classification, community detection, and subgraph-level prediction raise substantially harder protocol questions around leakage, overlapping supervision structures, and compatible splits, and we position these as non-trivial extensions requiring separate treatment.

Our work targets a complementary and more controlled setting: *same-graph* cross-task transfer between node classification and link prediction in *both directions*, with a unified protocol that standardizes the observed graph used for message passing, fixes node and edge splits, and fixes LP negatives across all methods. This enables apples-to-apples transfer comparisons and supports *multi-objective* evaluation when practitioners prefer a shared encoder for both tasks (e.g., to amortize training and serving cost and maintain a single embedding space).

**Evaluation protocols for node classification and link prediction.** Evaluation choices substantially affect conclusions in graph learning, and reported gains can be sensitive to split choice and experimental degrees of freedom (Lv et al., 2021). For node classification, early transductive benchmarks popularized fixed citation splits (Yang et al., 2016), while inductive protocols evaluate generalization to unseen nodes and are often paired with neighborhood sampling methods such as GraphSAGE (Hamilton et al., 2017). Dataset properties such as homophily and node distinguishability influence when message passing should help, motivating diagnostics that characterize regime behavior (Luan et al., 2023). Standardized benchmarks such as OGB improve reproducibility through curated datasets and official splits (Hu et al., 2020a).

For link prediction, protocol details are especially delicate: results depend on edge splits, negative sampling, and whether evaluation edges inadvertently appear in the adjacency used for message passing (Hu et al., 2020a). Prior work highlights that including validation or test edges in the observed graph can leak information and inflate performance, and motivates separating the representation-learning graph from evaluated edges (Zhang & Chen, 2018). Motivated by these issues, we propose a leakage-free protocol that fixes observed graphs, splits, and LP negatives, enabling fair bidirectional transfer studies between NC and LP on the same underlying graph.

## 3. Leakage-Free Evaluation Protocol

**Motivation and goal.** Many real-world graphs come with *multiple* supervised signals on the same underlying structure, most commonly node labels for node classification (NC) and observed edges for link prediction (LP). In practice, these objectives are often evaluated with task-specific

pipelines: NC models are trained and tested using only node supervision, while LP models are trained and tested using only edge supervision, frequently with different splits, different observed graphs, and different negative sampling. This makes it hard to answer a basic question: *when can supervision from one task be reused to improve the other on the same graph.* Naively combining tasks can also yield misleading performance gains, as evaluated edges or held-out labels may leak into node representations through message passing, and resampling LP negatives across methods can effectively alter the task being evaluated.

Our goal is therefore twofold: (i) define a leakage-free, standardized evaluation interface shared by NC and LP, and (ii) use it to measure cross-task transfer under fixed and reproducible conditions.

**Protocol requirements.** A valid cross-task comparison must satisfy three requirements: (i) **No edge leakage:** edges evaluated for LP must not appear in the graph used for message passing when computing embeddings, (ii) **Fixed negatives:** LP must use fixed negative sets for training, validation, and test, reused across all methods and runs, (iii) **Fixed splits:** all methods must use the same node split for NC supervision and the same edge split for LP positives. We summarize the protocol in the main text and defer implementation details and sanity checks to the appendix.

**Setup and notation.** Let $G = (V, E, X)$ be an undirected graph with node set $V$, edge set $E$, and node features $X \in \mathbb{R}^{|V| \times d}$. For NC, each node $v \in V$ may have a label $y_v \in \{1, \ldots, C\}$. For LP, the goal is to score candidate pairs $(u, v)$. We define:

- **Node split (NC supervision):** $V_{\mathrm{tr}}^{\mathrm{NC}}, V_{\mathrm{va}}^{\mathrm{NC}}, V_{\mathrm{te}}^{\mathrm{NC}}$.
- **Edge split (LP positives):** $E_{\mathrm{tr}}^{+}, E_{\mathrm{va}}^{+}, E_{\mathrm{te}}^{+}$ with $E$ the disjoint union of all three splits.
- **Fixed LP negatives:** $E_s^{-}$ for each $s \in \{\mathrm{tr}, \mathrm{va}, \mathrm{te}\}$.

In what follows, tr, va, and te denote training, validation, and test splits, respectively. All splits and negative sets are generated once per dataset with fixed seeds and reused across methods. We use $60\%/20\%/20\%$ for NC label splits and $80\%/10\%/10\%$ for LP positive edge splits, a standard choice that provides sufficient training edges for stable LP evaluation; the ratios are held constant across all experiments. We present the protocol for the transductive setting here; the inductive variant is detailed in Appendix A.4.

**What is given.** All nodes $V$ and features $X$ are available during training and evaluation. Only labels on $V_{\mathrm{tr}}^{\mathrm{NC}}$ are used to train NC, while labels on $V_{\mathrm{va}}^{\mathrm{NC}}$ and $V_{\mathrm{te}}^{\mathrm{NC}}$ are held out.

**LP positives and the observed message passing graph.** We split edges into LP positives $E_{\mathrm{tr}}^{+}, E_{\mathrm{va}}^{+}, E_{\mathrm{te}}^{+}$. To prevent

leakage of evaluated edges into representations, we define a single observed training adjacency

$$A_{\mathrm{obs}} := \mathrm{Adj}(V, E_{\mathrm{tr}}^+),$$

and require that *all* embeddings used for NC training, NC evaluation, and LP evaluation, including during transfer, are computed by message passing only on $A_{\mathrm{obs}}$. No method may include $E_{\mathrm{va}}^+$ or $E_{\mathrm{te}}^+$ in the message passing graph. This enforces a shared, leakage-free observed graph across tasks and methods.

**Fixed negatives for LP.** For each split $s \in \mathrm{tr, va, te}$, we generate a fixed negative set $E_s^- \subset (V \times V) \setminus E$ using a specified policy (reported with experiments). We use a 1:1 positive-to-negative ratio, so $|E_s^-| = |E_s^+|$. Negatives exclude self-loops and duplicates and do not overlap with any positive edge in $E$. All negative sets are generated once with fixed seeds and reused across methods and runs; LP evaluation uses candidate sets $(E_s^+, E_s^-)$.

**Summary.** This protocol fixes the supervision available in each split, fixes LP negatives, and enforces a strict separation between evaluated edges and the message passing graph. It provides a common, leakage-free interface for comparing NC, LP, and cross-task transfer on the same graph. For details and sanity checks (e.g., verifying no overlap between $A_{\mathrm{obs}}$ and evaluation positives, and no overlap between negatives and any positives), see Section A.3.

### 3.1. Combined performance metric

We evaluate each method on a given dataset with two task scores: node classification (NC) accuracy, denoted $s_N(\cdot)$, and link prediction (LP) AUC, denoted $s_L(\cdot)$. Since these metrics can have different scales and baseline levels across datasets, directly summing raw scores (e.g., $s_N(m) + s_L(m)$) is not meaningful and can obscure trade-offs. We therefore report a normalized combined metric, the *CoTask Score (CTS)*, based on dimensionless percent gains anchored to a fixed reference. The same construction applies if accuracy and AUC are replaced by other standard measures (e.g., macro-F1 for NC or Hits@K for LP).

**CoTask Score (CTS).** For cross-method comparison, we anchor both tasks to a fixed reference model $g$ computed once per dataset and reused for all methods; throughout, $g$ is a standard GCN trained and evaluated under our protocol. For any model $m$, define the task-wise relative gains over $g$:

$$\mathrm{RG}_N(m) := 100 \left( \frac{s_N(m)}{s_N(g)} - 1 \right),$$

$$\mathrm{RG}_L(m) := 100 \left( \frac{s_L(m)}{s_L(g)} - 1 \right).$$

We summarize a model's overall NC+LP performance with the CoTask Score

$$\mathrm{CTS}(m) := \tfrac{1}{2} \left( \mathrm{RG}_N(m) + \mathrm{RG}_L(m) \right). \quad (1)$$

$\mathrm{CTS}(m)$ is the average percent gain over $g$ across accuracy and AUC, yielding a single interpretable, unitless measure of overall improvement while preserving the relative scaling of each task.

## 4. Cross-Task Transfer Methods

**Motivation and scope.** Many real graphs come with *both* node labels (NC supervision) and observed edges (LP supervision), yet standard pipelines typically train NC models without explicitly leveraging edge-level supervision and train LP models without leveraging node labels. We study when and how supervision from one task can improve the other on the *same* graph, under the leakage-free protocol described in Section 3. Our focus is on *transfer regimes*, not new architectures.

**Shared parameter-space viewpoint.** For a fixed backbone architecture and protocol graph, training defines an optimization problem over the *model parameters* (equivalently, over the induced function class). Concretely, the same encoder family can be used for both tasks as a mapping $E_\theta : \mathbb{R}^{|V| \times d} \to \mathbb{R}^{|V| \times p}$, producing node representations that are then scored by a task head. NC training seeks parameters $\theta_{\mathrm{NC}} \in \arg\min_\theta \mathcal{L}_{\mathrm{NC}}(\theta)$, while LP training seeks $\theta_{\mathrm{LP}} \in \arg\min_\theta \mathcal{L}_{\mathrm{LP}}(\theta)$, where both losses are evaluated under the same leakage-free protocol in Section 3. Because $\mathcal{L}_{\mathrm{NC}}$ and $\mathcal{L}_{\mathrm{LP}}$ define different landscapes over the *same* parameter space (Figure 1), their minimizers can be close, compatible, or conflicting. This motivates our transfer regimes as controlled ways to navigate these landscapes: WS initializes the target optimization near a source-task solution, ET reuses a source representation as an input signal while relearning parameters for the target task, and MV/Joint explicitly couple objectives during training to bias optimization toward parameters that perform well on both tasks. We provide additional geometric intuition in the appendix. For additional intuition on why cross-task transfer can help or hurt, Appendix A.5 discusses a shared parameter-space and shared latent space view of NC and LP optimization under a fixed backbone.

**Common setup.** All methods share a backbone encoder $E$ (GCN, GraphSAGE, GPS) that maps nodes to embeddings $Z = E(X, A) \in \mathbb{R}^{|V| \times d}$ using the protocol message-passing adjacency $A$. All regimes compute embeddings on the same protocol adjacency $A$ and train/evaluate LP on the same fixed candidate sets $(E_s^+, E_s^-)$ (fixed negatives) defined in Section 3. We write the NC model as $F = (E_F, C)$ with encoder $E_F$ and classifier head $C$, and the LP model as $H = (E_H, P)$ with encoder $E_H$ and link predictor $P$.

**Task losses and LP predictor.** Let $\mathcal{L}_{\mathrm{NC}}$ be cross-entropy over labeled nodes and $\mathcal{L}_{\mathrm{LP}}$ be binary cross-entropy over candidate edge sets $(E_s^+, E_s^-)$ for split $s$. For LP, given

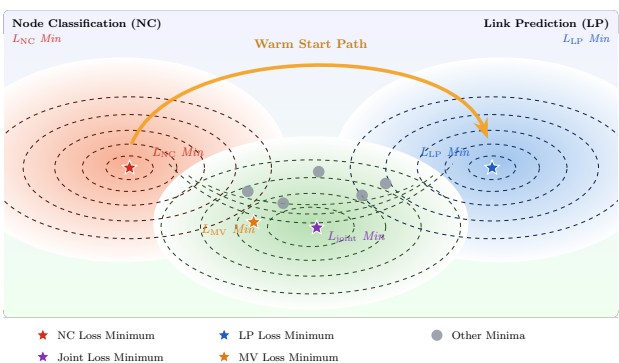

*Figure 1.* **Cross-task transfer as optimization in a shared parameter space.** For a fixed GNN/GT architecture, NC and LP correspond to different losses over the same parameters, yielding distinct (possibly incompatible) minima. Warm start initializes the target training near a source-task solution, while MV and Joint couple objectives to steer optimization toward solutions that perform well on both tasks; other local minima illustrate potential negative transfer.

node embeddings $z_u, z_v$, we form a pairwise feature vector

$$\phi(z_u, z_v) := [\, z_u \,\|\, z_v \,\|\, |z_u - z_v| \,\|\, (z_u \odot z_v)\,]\,,$$

and predict $\hat{y}_{uv} = P(\phi(z_u, z_v))$ with a MLP $P$.

### 4.1. Warm start (WS)

Warm start transfers *parameters* by initializing the target encoder with the encoder learned on the source task.

**Procedure.** Train the source model (NC or LP) to obtain encoder weights $\theta_{\mathrm{src}}$. Initialize the target encoder with these weights and train the target task with a fresh task head $h_{\mathrm{tgt}}$ (i.e., $C$ for NC or $P$ for LP), randomly re-initialized:

$$\theta_{\mathrm{tgt}} \leftarrow \theta_{\mathrm{src}}, \qquad h_{\mathrm{tgt}} \sim \text{random init,}$$

then optimize the target loss on the target supervision. For NC $\rightarrow$ LP, we set $E_H \leftarrow E_F$ at initialization and train $(E_H, P)$ on LP; LP $\rightarrow$ NC is defined analogously.

### 4.2. Embedding transfer (ET)

Embedding transfer treats the source representation as a reusable signal and injects it into the target model via modified node inputs. We compute frozen source embeddings once and then train the target model using a transformed feature matrix. In ET regimes we train a *full target encoder from scratch* on the target supervision (not only a linear probe), with the transfer signal entering solely through the input features.

**Frozen source embeddings and projection.** After training the source encoder, compute $Z_{\mathrm{src}} := E_{\mathrm{src}}(X, A) \in \mathbb{R}^{|V| \times d_{\mathrm{src}}}$, and stop gradients through $Z_{\mathrm{src}}$ during target training. If $d_{\mathrm{src}}$ does not match the target input dimension, we apply a linear projection $\Pi$ and use $\tilde{Z}_{\mathrm{src}} := \Pi(Z_{\mathrm{src}})$. We

treat $\Pi$ as part of the source model (trained on the source task and then frozen together with $E_{\mathrm{src}}$).

◇ **ET Replace (ET Rep).** Replace the original node features by the transferred embeddings: $X' := \tilde{Z}_{\mathrm{src}}$. Train a new target model from scratch using $X'$ as input, optimizing $(E_{\mathrm{tgt}}, \mathrm{head}_{\mathrm{tgt}})$ on the target supervision. ET Rep tests whether the source representation is sufficient for the target task when used as the sole input signal.

◇ **ET Concat.** Concatenate original features with transferred embeddings: $X' := [\, X \,\|\, \tilde{Z}_{\mathrm{src}}\,]$. Train a new target model from scratch on the target task using $X'$. ET Concat retains access to raw features while leveraging complementary information present in the source representation.

### 4.3. Multi-view (MV)

Multi-view learning trains two task-specific encoders jointly on the same protocol graph while explicitly aligning their representations. This regime measures *simultaneous learning with representational compatibility*.

**Views and supervision.** We maintain two encoders and heads: $z_v^F = E_F(X, A)_v$, $z_v^H = E_H(X, A)_v$, with standard supervised losses $\mathcal{L}_{\mathrm{NC}}$ (via head $C$) and $\mathcal{L}_{\mathrm{LP}}$ (via head $P$). To encourage compatibility between NC and LP representations, we add a cross-view alignment term on a set of training nodes $\mathcal{V}_{\mathrm{align}}$ that never uses held-out supervision. In transductive experiments we take $\mathcal{V}_{\mathrm{align}} = V_{\mathrm{tr}}^{\mathrm{NC}}$, and in inductive experiments we take $\mathcal{V}_{\mathrm{align}} = V_{\mathrm{tr}}$.

**Cross-view alignment (InfoNCE with in-batch negatives).** Let $\pi_F, \pi_H$ be small projection MLPs and $\mathrm{sim}(\cdot, \cdot)$ be cosine similarity. For a minibatch $\mathcal{B} \subseteq \mathcal{V}_{\mathrm{align}}$, define

$$\ell_{F \rightarrow H}(\mathcal{B}) = -\sum_{v \in \mathcal{B}} \log \frac{\exp \frac{\mathrm{sim}\left(\pi_F\left(z_v^F\right), \pi_H\left(z_v^H\right)\right)}{\tau}}{\sum_{u \in \mathcal{B}} \exp \frac{\mathrm{sim}\left(\pi_F\left(z_v^F\right), \pi_H\left(z_u^H\right)\right)}{\tau}}$$

We symmetrize the objective by swapping views and set

$$\mathcal{L}_{\mathrm{x}}(\mathcal{B}) := \tfrac{1}{2}\left(\ell_{F \rightarrow H}(\mathcal{B}) + \ell_{H \rightarrow F}(\mathcal{B})\right),$$

so negatives are provided by other nodes in the same minibatch (in-batch negatives).

**MV training loss.** The multi-view objective is $\mathcal{L}_{\mathrm{MV}} = \mathcal{L}_{\mathrm{NC}} + \mathcal{L}_{\mathrm{LP}} + \lambda_{\mathrm{x}} \mathcal{L}_{\mathrm{x}}$, where $\lambda_{\mathrm{x}}$ controls the strength of cross-view alignment and $\tau$ is the temperature. We keep unit weights on $\mathcal{L}_{\mathrm{NC}}$ and $\mathcal{L}_{\mathrm{LP}}$ to reduce degrees of freedom and isolate the effect of representation alignment; only $\lambda_{\mathrm{x}}$ is tuned on validation. Both supervised losses are averaged per example to prevent scale imbalance. After training, MV yields two aligned encoders; for transfer, we use the corresponding view encoder for initialization or embedding reuse.

## 4.4. Joint training (Joint)

Joint training uses a single shared encoder optimized simultaneously for NC and LP with task-specific heads. Unlike WS and ET, which transfer *after* training a source task, Joint induces transfer *during* training through shared representation learning.

**Model.** We use one encoder $E$ to produce embeddings $z_v = E(X, A)_v$ and attach two heads, an NC head $C$ and an LP head $P$.

**Training loss.** We optimize a weighted sum of the task losses: $\mathcal{L}_{\text{Joint}} = \lambda \mathcal{L}_{\text{NC}} + (1 - \lambda) \mathcal{L}_{\text{LP}}$, with $\lambda$ selected on the validation split via a small fixed grid.

**Early stopping and checkpoint selection.** Because accuracy and AUC have different scales, we select checkpoints using a dimensionless combined validation criterion based on relative improvements over the corresponding single-task baselines:

$$S_{\text{val}} = \frac{1}{2} \left( \frac{\text{Acc}_{\text{NC}}^{\text{va}}}{\text{Acc}_{\text{NC,base}}^{\text{va}}} + \frac{\text{AUC}_{\text{LP}}^{\text{va}}}{\text{AUC}_{\text{LP,base}}^{\text{va}}} \right).$$

Note that $S_{\text{val}}$ is used only for checkpoint selection and is anchored to task-specific single-task baselines, whereas CTS (Section 3.1) is a reporting metric anchored to a fixed reference model. We use the same $S_{\text{val}}$ form for any regime that requires a combined validation signal (details in the appendix).

**Discussion.** WS isolates encoder parameter reuse, ET Rep and ET Concat isolate representation reuse via input injection, MV learns two task-specific encoders with explicit alignment, and Joint learns a single shared representation under both objectives. Together, these regimes span a controlled spectrum of cross-task transfer mechanisms for analyzing when and why NC and LP help each other under a leakage-free protocol.

## 5. Experiments

We evaluate same-graph cross-task transfer under the leakage-free protocol in Section 3. Our goals are to (i) quantify bidirectional NC-LP transfer, (ii) test robustness across backbones and transductive/inductive settings, and (iii) identify simple diagnostics predicting when transfer helps.

**Homophily measures.** We use two standard homophily measures to characterize each dataset. *Edge homophily $H_e$* is the fraction of edges that connect nodes sharing the same label, and *node homophily $H_n$* is the average over nodes of the fraction of same-label neighbors. Formal defintions of $H_e$ and $H_n$ can be found in Appendix A.2. Both measures range in $[0, 1]$, with values near 1 indicating strong label-connectivity alignment (homophilic graphs) and values near

0 indicating that edges predominantly connect nodes of different labels (heterophilic graphs).

**Datasets.** We consider 11 commonly used node-level benchmarks spanning homophilic, heterophilic, and mixed (structure-dominant) regimes. Homophilic citation graphs include CORA, CITESEER, and PUBMED. Heterophilic datasets include TEXAS, WISCONSIN, CORNELL, ACTOR, and ROMAN-EMPIRE. Finally, USA, EUROPE, and BRAZIL exhibit mixed homophily but share a characteristic structure-dominant signature: LP is highly learnable while NC has substantial headroom. We operationalize this regime as follows: under a fixed backbone and protocol, a dataset is structure-dominant if its baseline LP AUC ranks in the upper half of the benchmark while its baseline NC accuracy ranks in the lower half. This criterion is threshold-free and reproducible given a fixed dataset collection and evaluation protocol, and avoids imposing an arbitrary absolute cutoff on metrics that are not directly comparable across datasets and backbones. Empirically, USA, EUROPE, and BRAZIL satisfy this criterion across all three backbones under our protocol, while homophilic and heterophilic datasets do not. In this regime, LP supervision acts as structural pretraining for NC, improving accuracy even when homophily is low. For each dataset, we report edge- and node-level homophily $(H_e, H_n)$ and global clustering coefficient (Global CC) as lightweight diagnostics (Table 1).

*Table 1.* **Dataset statistics.** We report graph size, feature dimensionality, and class count, together with edge homophily $H_e$, node homophily $H_n$, and global clustering coefficient (CC).

| DATASET | NODES | EDGES | FEAT | CLASS | $H_e$ | $H_n$ | CC |
|---|---|---|---|---|---|---|---|
| CORA | 2,708 | 5,429 | 1,433 | 7 | 0.81 | 0.83 | 0.09 |
| CITESEER | 3,327 | 4,732 | 3,703 | 6 | 0.74 | 0.71 | 0.13 |
| PUBMED | 19,717 | 44,338 | 500 | 3 | 0.80 | 0.79 | 0.05 |
| TEXAS | 183 | 309 | 1,703 | 5 | 0.11 | 0.07 | 0.03 |
| CORNELL | 183 | 295 | 1,703 | 5 | 0.13 | 0.11 | 0.04 |
| WISCONSIN | 251 | 499 | 1,703 | 5 | 0.20 | 0.17 | 0.04 |
| ACTOR | 7,600 | 33,544 | 931 | 5 | 0.22 | 0.22 | 0.02 |
| ROMAN | 22,662 | 32,927 | 300 | 18 | 0.05 | 0.05 | 0.29 |
| USA | 1,190 | 28,288 | – | 4 | 0.70 | 0.37 | 0.43 |
| EUROPE | 399 | 12,385 | – | 4 | 0.45 | 0.27 | 0.33 |
| BRAZIL | 131 | 2,137 | – | 4 | 0.41 | 0.22 | 0.45 |

**Backbones and heads.** All methods use the same backbone encoder family $E \in \{\text{GCN}, \text{GraphSAGE}, \text{GPS}\}$ and differ only in how source-task supervision is reused. We include GCN and GraphSAGE as canonical message-passing baselines and GPS as a competitive graph transformer backbone, spanning message-passing and attention-based families. For NC, we use a linear classifier head. For LP, we use an MLP predictor over standard pairwise features of node embeddings (concatenation and elementwise product). Unless noted otherwise, hidden dimension and depth are fixed per backbone and tuned once on validation, then reused across all transfer regimes to reduce degrees of freedom.

**Transfer regimes.** We evaluate five transfer regimes from Section 4 in both directions, NC $\rightarrow$ LP and LP $\rightarrow$ NC: warm start (WS), embedding transfer by replacement (ET Rep), embedding transfer by concatenation (ET Concat), Multi-view (MV), and Joint training (Joint). In all cases, embeddings are computed on the protocol adjacency matrix $A$, and LP training and evaluation use the fixed candidate sets $(E_s^+, E_s^-)$ defined in Section 3. We report improvements relative to the corresponding single-task base model trained under the same protocol.

**Settings and evaluation.** We report transductive results (shared node set, fixed edge splits) for all datasets. All models use a two-layer GNN encoder with fixed hidden dimension and dropout, learning rate is fixed at 0.01 and training has a maximum of 200 epochs with early stopping based on validation performance. NC uses cross-entropy loss and we report accuracy; LP uses binary cross-entropy loss and we report ROC-AUC. In addition to per-task metrics, we report the CoTask Score (CTS) from Section 3.1 as a compact summary, while always including the underlying NC and LP scores in the main tables.

**Implementation details.** All splits and fixed negative sets are generated once per dataset with fixed random seeds and reused for all methods. For each dataset and backbone, we tune learning rate, weight decay, dropout, and early-stopping patience on the validation split, and then reuse the same tuned hyperparameters across transfer regimes to keep comparisons controlled. For Joint training, we tune the task-weight $\lambda$ in $\mathcal{L}_{\text{Joint}} = \lambda\mathcal{L}_{\text{NC}} + (1-\lambda)\mathcal{L}_{\text{LP}}$ on validation using the combined criterion $S_{\text{val}}$ defined in Section 4.4. For MV, we keep unit weights on $\mathcal{L}_{\text{NC}}$ and $\mathcal{L}_{\text{LP}}$ and tune only the alignment weight $\lambda_{\text{x}}$ in $\mathcal{L}_{\text{MV}} = \mathcal{L}_{\text{NC}} + \mathcal{L}_{\text{LP}} + \lambda_{\text{x}}\mathcal{L}_{\text{x}}$ (see Section A.1). The appendix reports full hyperparameter grids, negative sampling policies, and sanity checks for leakage (e.g., verifying $E_{\text{va}}^+ \cup E_{\text{te}}^+$ does not appear in $A_{\text{obs}}$, and that negatives do not overlap with any positives).

## 5.1. Main transfer results across backbones

Tables 2 and 3 summarize same-graph cross-task transfer in both directions, with GCN, GraphSAGE, and GPS reported side by side. The key message is that **transfer is directional and regime dependent**: NC supervision reliably improves LP on homophilic graphs, whereas LP supervision improves NC mainly in mixed, structure-dominant settings (high LP learnability with remaining headroom in NC). Overall, NC$\rightarrow$LP behaves like label-aligned representation reuse, while LP$\rightarrow$NC behaves like structure-driven pretraining that helps only in the right regime. The results for inductive setting are given in Section A.4.

**NC$\rightarrow$LP is robust on homophilic graphs.** On CORA, CITESEER, and PUBMED, *every* backbone achieves consistent AUC gains from NC$\rightarrow$LP across multiple mechanisms.

Moreover, methods that explicitly combine node supervision with graph structure during training (MV and Joint) often outperform warm-start style transfer, indicating that the benefit is not merely optimization but a representational effect. This supports a simple mechanism: when edges connect same-label nodes, NC training shapes embeddings whose local neighborhoods already encode the right inductive bias for edge scoring. Joint uses the same encoder capacity as the single-task baselines (only adding a lightweight second head), so its gains are not explained by increased model size. MV does train two task-specific encoders, but each task is evaluated using its corresponding encoder (we do not ensemble or expand the backbone at test time); thus improvements reflect coupled training and alignment rather than simply evaluating a higher-capacity model.

**LP$\rightarrow$NC is weaker and can incur negative transfer.** In homophilic graphs, the reverse direction is less reliable: gains are often small, and ET Rep can substantially *decrease* accuracy, indicating that LP optimization does not generally produce node-discriminative features under our leakage-free protocol. Thus, *pretrain on LP then reuse* is not a safe default, even when NC$\rightarrow$LP works well.

**When LP$\rightarrow$NC works, it looks like structural pretraining.** The clearest positive LP$\rightarrow$NC effects concentrate on USA, EUROPE, and BRAZIL, where LP is highly learnable while NC has substantial headroom. Here, ET Concat and Joint often provide the strongest improvements (and ET Rep can also help), consistent with the view that LP supervision supplies transferable *structural* features that can be repurposed for node labels when labels are not the primary organizing principle of connectivity.

**Heterophily alone does not determine success, but coupling objectives is safest.** On heterophilic datasets (TEXAS, CORNELL, WISCONSIN, ACTOR, ROMAN-EMPIRE), outcomes vary by dataset and backbone, and no single transfer mechanism dominates uniformly. However, a consistent practical pattern is that MV and Joint are the most stable choices, while ET Rep exhibits the largest variance, ranging from strong gains to severe negative transfer. This motivates predictors beyond homophily for LP$\rightarrow$NC and highlights mechanism choice as essential for avoiding negative transfer. These trends are consistent across GCN, GraphSAGE, and GPS, indicating that the asymmetry is not an artifact of a particular backbone.

## 5.2. Combined performance via CoTask Score

Directional tables isolate *where* transfer helps, but overall utility depends on whether gains in one task offset losses in the other. To capture this tradeoff, we report the CoTask Score (CTS), which averages percent improvements over a fixed GCN reference across NC and LP. Table 5 provides the full per-dataset CTS results (ordered by homophily regime),

*Table 2.* **NC→LP transfer results (AUC).** We report LP transfer gains (in percentage points) from NC-driven transfer, measured relative to the corresponding LP single-task base model. For each dataset and backbone, we bold the largest gain.

| DATASET | $H_e$ | $H_n$ | CC | GCN | | | | | | GraphSAGE | | | | | | GPS | | | | | |
|---|---|---|---|---|---|---|---|---|---|---|---|---|---|---|---|---|---|---|---|---|---|
| | | | | BASE | WS | ET REP | ET CON | MV | JOINT | BASE | WS | ET REP | ET CON | MV | JOINT | BASE | WS | ET REP | ET CON | MV | JOINT |
| CORA | 0.81 | 0.83 | 0.09 | 80.3 | 9.4 | 10.7 | 9.8 | 10.1 | **12.3** | 76.5 | 11.2 | 10.2 | 11.0 | 13.0 | **13.3** | 77.2 | 9.0 | 12.7 | 10.8 | **14.2** | 13.6 |
| CITESEER | 0.74 | 0.71 | 0.13 | 77.0 | 11.9 | 11.9 | 11.1 | 12.8 | **13.2** | 75.0 | 11.0 | 9.3 | 9.4 | 12.4 | **13.4** | 77.4 | 5.6 | 11.3 | 8.6 | **13.4** | 11.0 |
| PUBMED | 0.80 | 0.79 | 0.05 | 89.8 | **5.9** | 4.1 | 4.5 | 5.0 | 4.7 | 84.8 | 1.5 | -4.3 | -0.7 | 3.9 | **4.5** | 93.6 | -0.2 | -0.6 | 0.4 | **2.4** | 2.3 |
| TEXAS | 0.11 | 0.07 | 0.03 | 67.0 | 3.6 | -0.3 | -2.6 | 4.7 | **7.1** | 73.7 | 2.0 | 3.3 | 4.0 | 6.4 | **6.8** | 73.7 | 2.9 | 2.0 | 0.7 | 7.6 | **8.9** |
| CORNELL | 0.13 | 0.11 | 0.04 | 75.4 | **4.6** | -2.1 | -1.3 | 3.7 | 2.3 | 79.2 | 1.2 | 0.9 | 1.2 | **1.7** | 0.4 | 79.8 | -2.7 | 0.0 | -1.4 | -0.1 | **0.1** |
| WISCONSIN | 0.20 | 0.17 | 0.04 | 75.1 | 1.8 | -0.2 | -0.4 | 2.3 | **3.7** | 75.2 | 3.1 | 5.8 | 6.6 | **9.0** | 7.1 | 73.1 | 2.9 | 4.6 | 5.8 | **9.4** | 8.9 |
| ACTOR | 0.22 | 0.22 | 0.02 | 80.4 | -0.3 | -1.6 | -0.7 | -0.3 | **1.2** | 79.8 | **0.0** | -3.4 | -0.0 | -2.7 | -0.5 | 78.7 | 0.9 | -0.2 | -1.2 | 0.4 | **2.0** |
| ROMAN | 0.05 | 0.05 | 0.29 | 73.8 | **0.1** | -10.6 | -6.6 | -0.1 | -11.1 | 63.5 | -0.1 | -12.5 | -13.1 | -2.5 | **1.2** | 68.8 | -5.4 | -7.3 | -6.1 | -6.8 | **27.9** |
| USA | 0.70 | 0.37 | 0.43 | 95.5 | **0.3** | 0.1 | 0.3 | 0.3 | -0.6 | 95.2 | 0.0 | -0.3 | **0.5** | 0.0 | 0.3 | 94.8 | 0.5 | **0.9** | 0.7 | 0.3 | 0.1 |
| EUROPE | 0.45 | 0.27 | 0.33 | 92.6 | **0.4** | -1.1 | 0.1 | 0.2 | -2.3 | 92.0 | -0.0 | -1.5 | **0.4** | -1.1 | -1.9 | 91.6 | -0.4 | 0.6 | **0.8** | -0.8 | -1.7 |
| BRAZIL | 0.41 | 0.22 | 0.45 | 90.4 | **3.1** | 0.1 | 0.9 | 1.3 | 0.1 | 90.1 | 1.1 | 0.3 | **1.4** | 0.9 | 0.4 | 90.5 | -0.7 | 0.2 | **0.5** | -0.5 | -0.2 |

*Table 3.* **LP→NC transfer results (Acc).** We report NC transfer gains (in percentage points) from LP-driven transfer, measured relative to the corresponding LP single-task base model. For each dataset and backbone, we bold the largest gain.

| DATASET | $H_e$ | $H_n$ | CC | GCN | | | | | | GraphSAGE | | | | | | GPS | | | | | |
|---|---|---|---|---|---|---|---|---|---|---|---|---|---|---|---|---|---|---|---|---|---|
| | | | | BASE | WS | ET REP | ET CON | MV | JOINT | BASE | WS | ET REP | ET CON | MV | JOINT | BASE | WS | ET REP | ET CON | MV | JOINT |
| CORA | 0.81 | 0.83 | 0.09 | 85.3 | **0.6** | -12.7 | 0.5 | 0.5 | 0.4 | 86.0 | -0.2 | -18.7 | 0.0 | **0.6** | -0.2 | 78.4 | 0.0 | -18.3 | 1.4 | **4.7** | 4.5 |
| CITESEER | 0.74 | 0.71 | 0.13 | 71.6 | 0.5 | -16.8 | 0.1 | 0.3 | **3.5** | 74.2 | 0.3 | -25.3 | -0.1 | **0.5** | -0.3 | 68.6 | -5.2 | -27.8 | -1.4 | 3.3 | **6.3** |
| PUBMED | 0.80 | 0.79 | 0.05 | 88.5 | **0.1** | -21.4 | -1.0 | -0.1 | -0.5 | 88.8 | 0.1 | -20.4 | -0.6 | -0.1 | **0.3** | 86.6 | -2.0 | -23.1 | -14.7 | **1.7** | 1.3 |
| TEXAS | 0.11 | 0.07 | 0.03 | 50.3 | 2.6 | 6.6 | 1.6 | -0.3 | **11.9** | 82.9 | 0.0 | -13.2 | -5.0 | **1.3** | -2.4 | 56.6 | 9.2 | 6.8 | 7.6 | -6.1 | **20.5** |
| CORNELL | 0.13 | 0.11 | 0.04 | 51.6 | **-0.3** | -11.1 | -2.9 | -2.4 | -2.6 | 72.6 | **0.5** | -21.3 | -7.4 | -2.6 | -5.0 | 49.5 | 10.0 | -8.4 | 7.1 | 7.4 | **15.3** |
| WISCONSIN | 0.20 | 0.17 | 0.04 | 49.4 | -0.2 | -2.9 | -1.6 | -0.8 | **1.0** | 80.0 | **-1.0** | -22.9 | -8.8 | -1.8 | -3.3 | 64.5 | -12.2 | -13.9 | 2.6 | 2.0 | **15.7** |
| ACTOR | 0.22 | 0.22 | 0.02 | 27.9 | 0.5 | -1.3 | 1.0 | 0.7 | **3.2** | 34.2 | -0.2 | -7.6 | 0.3 | **0.4** | 0.1 | 33.1 | -0.4 | -4.5 | -2.2 | **1.5** | 0.0 |
| ROMAN | 0.05 | 0.05 | 0.29 | 51.4 | **0.2** | -28.2 | -0.4 | -0.0 | -0.5 | 74.6 | **0.2** | -27.5 | -0.8 | -0.9 | -1.4 | 75.4 | -1.9 | -34.5 | 0.4 | -0.3 | **0.5** |
| USA | 0.70 | 0.37 | 0.43 | 54.6 | -1.1 | 5.8 | **7.8** | -0.2 | 6.2 | 55.8 | -1.7 | 3.2 | **5.2** | -1.5 | 4.7 | 34.5 | 2.4 | **24.2** | 17.2 | 12.3 | 20.5 |
| EUROPE | 0.45 | 0.27 | 0.33 | 53.7 | -0.4 | 2.6 | 1.4 | -1.0 | **2.8** | 35.2 | 4.9 | 16.0 | **17.9** | 3.6 | 15.7 | 37.7 | -2.1 | 11.7 | **12.6** | 0.5 | 7.2 |
| BRAZIL | 0.41 | 0.22 | 0.45 | 48.9 | -0.4 | **15.2** | 6.7 | 2.2 | 13.7 | 36.7 | 5.2 | **20.4** | 17.8 | 0.4 | 15.6 | 39.6 | 3.7 | 16.7 | **22.6** | 2.2 | 10.7 |

while Table 8 in the appendix aggregates CTS by regime for a compact summary. CTS sharpens the same story in a single number. On homophilic graphs, the strongest CTS values are typically achieved by MV and Joint, confirming that NC-informed transfer yields broad improvements when neighborhoods are label coherent. On heterophilic graphs, positive CTS concentrates in the stronger backbones (GraphSAGE and especially GPS) and again favors MV or Joint, indicating that coupled objectives are more reliable than post hoc reuse when homophily is low. Finally, the mixed regime shows that combined gains are selective rather than automatic: some mechanisms deliver meaningful net improvements, while others degrade one task enough to erase progress on the other. Overall, CTS reinforces that same-graph transfer is not a one-size-fits-all recipe, and that both dataset regime and mechanism choice govern whether transfer produces net benefit.

**Homophily predicts NC→LP gains, but not LP→NC.** Table 4 links the observed asymmetry to dataset statistics under a GCN encoder. For NC→LP, gains correlate positively with homophily across mechanisms, and the dependence is strongest for representation-fusion approaches (ET Concat and ET Rep), consistent with the idea that label-coherent neighborhoods make node-supervised embeddings directly reusable for edge scoring. For LP→NC, correlations with $H_e$ and $H_n$ are weak and inconsistent, indicating that link

supervision does not improve node accuracy in a homophily-driven way. Instead, Global clustering coefficient can matter for specific mechanisms (notably ET Concat), suggesting a distinct driver tied to transitivity rather than label alignment. We give correlations for all backbones in Table 7.

*Table 4.* **Homophily predicts NC→LP gains more consistently than LP→NC gains (GCN, Pearson).** Entries report Pearson correlation $r$ between transfer gains and dataset statistics across 11 datasets. For LP→NC and NC→LP, we correlate the gains (relative to the base model) with the same statistics. Stars indicate two-sided significance tests for zero correlation: $^{*}p < 0.05$, $^{**}p < 0.01$, $^{***}p < 0.001$. $H_e$ and $H_n$ denote edge- and node-level homophily, and Global CC denotes global clustering coefficient.

| METHOD | LP → NC | | | NC → LP | | |
|---|---|---|---|---|---|---|
| | $H_e$ | $H_n$ | GLOBAL CC | $H_e$ | $H_n$ | GLOBAL CC |
| WS | -0.275 | -0.086 | -0.548 | 0.584 | **0.736**$^{**}$ | -0.336 |
| ET REP | -0.114 | -0.353 | 0.370 | **0.787**$^{**}$ | **0.847**$^{***}$ | -0.249 |
| ET CONCAT | 0.270 | -0.036 | **0.782**$^{**}$ | **0.838**$^{**}$ | **0.904**$^{***}$ | -0.168 |
| MV | 0.279 | 0.239 | 0.380 | 0.548 | **0.709**$^{*}$ | -0.394 |
| JOINT | -0.059 | -0.241 | 0.428 | 0.532 | **0.650**$^{*}$ | -0.510 |

**Takeaways.** Across datasets and backbones, same-graph cross-task transfer is strongly directional and regime dependent: NC→LP gains are reliably positive on homophilic graphs and are well predicted by homophily (especially for ET Concat and ET Rep), whereas LP→NC gains are not explained by homophily and instead concentrate in structure-dominant regimes where LP is highly learnable but NC has headroom. As a result, transfer should not be treated as symmetric by default, and both mechanism choice and regime awareness are essential to avoid negative transfer.

*Table 5.* **Combined performance.** CoTask Score (CTS) summarizes joint NC+LP performance as the average percent improvement over a fixed GCN reference across the two tasks. For each dataset, we highlight the **best**, **second**, and **third** CTS values.

| DATASET | GCN BACKBONE | | | | | | GraphSAGE BACKBONE | | | | | | GPS BACKBONE | | | | | |
|---|---|---|---|---|---|---|---|---|---|---|---|---|---|---|---|---|---|---|
| | BASE | WS | ET REP | ET CON | MV | JOINT | BASE | WS | ET REP | ET CON | MV | JOINT | BASE | WS | ET REP | ET CON | MV | JOINT |
| CORA | 0.0 | 6.2 | -0.8 | 6.4 | 8.0 | 7.8 | -1.9 | 4.9 | -6.5 | 4.9 | 6.6 | 6.3 | -6.3 | -0.4 | -8.8 | 1.6 | 5.6 | 5.2 |
| CITESEER | 0.0 | 8.0 | -4.0 | 7.3 | 10.9 | 11.2 | 0.5 | 7.8 | -11.1 | 6.6 | 8.9 | 9.0 | -1.9 | -1.8 | -13.9 | 2.7 | 9.2 | 9.7 |
| PUBMED | 0.0 | 3.3 | -9.8 | 1.9 | 2.4 | 2.9 | -2.6 | -1.7 | -16.5 | -3.3 | -0.5 | 0.0 | 1.0 | -0.2 | -12.4 | -7.1 | 3.3 | 3.1 |
| TEXAS | 0.0 | 5.3 | 6.4 | -0.4 | 4.6 | 17.1 | 24.7 | 39.0 | 26.9 | 35.5 | 43.6 | 40.3 | 10.6 | 22.6 | 19.6 | 19.5 | 11.0 | 38.4 |
| CORNELL | 0.0 | 2.8 | -12.1 | -3.7 | 0.8 | -1.2 | 17.0 | 24.3 | 2.8 | 16.6 | 21.5 | 18.4 | 0.8 | 8.8 | -7.3 | 6.8 | 7.9 | 15.7 |
| WISCONSIN | 0.0 | 1.0 | -3.1 | -1.9 | 0.6 | 3.5 | 19.2 | 32.1 | 11.7 | 26.5 | 35.3 | 32.4 | 10.4 | 3.6 | 3.0 | 20.4 | 22.2 | 35.7 |
| ACTOR | 0.0 | 0.6 | -3.3 | 1.3 | 2.7 | 6.4 | 8.8 | 10.4 | -4.9 | 11.4 | 9.9 | 10.7 | 6.8 | 8.1 | 0.2 | 3.6 | 11.2 | 9.5 |
| ROMAN | 0.0 | 0.2 | -34.6 | -4.8 | -7.8 | -7.9 | 8.6 | 15.8 | -19.6 | 6.0 | 13.0 | 15.0 | 12.5 | 14.4 | -18.6 | 16.2 | 15.0 | 39.3 |
| USA | 0.0 | -0.8 | 5.4 | 7.3 | -0.3 | 5.4 | 1.0 | -0.5 | 3.8 | 6.1 | -0.4 | 5.4 | -29.5 | -16.4 | 3.9 | -2.6 | -7.3 | 0.1 |
| EUROPE | 0.0 | -0.1 | 1.8 | 1.3 | -2.2 | 1.4 | -26.6 | -12.9 | -3.4 | -0.7 | -14.8 | -3.9 | -21.8 | -17.6 | -4.2 | -3.3 | -15.4 | -9.8 |
| BRAZIL | 0.0 | 1.3 | 15.6 | 7.3 | -0.8 | 14.1 | -16.8 | -6.8 | 8.3 | 6.3 | -11.8 | 3.4 | -11.6 | -6.0 | 7.7 | 13.9 | -7.5 | 1.4 |

# 6. Conclusion

We studied same-graph cross-task transfer between node classification (NC) and link prediction (LP) under a leakage-free protocol that fixes node and edge splits, uses fixed LP negatives, and excludes evaluated edges from the message-passing graph. Across datasets, backbones, and transfer mechanisms, we find a consistent directional asymmetry: NC→LP is reliably beneficial on homophilic graphs, while LP→NC is less predictable, can induce negative transfer, and is most effective in structure-dominant settings where LP is easy but NC remains unsaturated. CoTask Score further shows that coupled training (MV, Joint) is typically more stable than post hoc reuse, and correlation analyses suggest homophily explains NC→LP gains more consistently than LP→NC, pointing to distinct drivers across directions. We hope these protocols, metrics, and findings provide a reproducible basis for deciding when and how to reuse supervision across objectives on the same graph.

**Scope and future directions.** Several natural extensions remain open: cross-graph and cross-domain settings where transfer must hold under distribution shift; whether the directional asymmetry persists under newer long-range architectures such as state-space graph models; extending the protocol to other task pairs such as community detection or graph-level objectives; and adaptive transfer frameworks that dynamically adjust task coupling during training via gradient-based loss balancing.

## Software and Data

Our code can be found at https://github.com/ avp-neelam/CrossTaskTransfer, all datasets used can be found through PyTorch Geometric loaders.

## Acknowledgements

This work was partially supported by the National Science Foundation under grants DMS-2220613, DMS-2229417, DMS-2204795, OAC-2115094, CNS-2331424, and ITE-2452833; ARL/Army Research Office awards W911NF-24-1-0202 and W911NF-24-2-0114; and Virginia Commonwealth Cyber Initiative grants. The authors acknowledge the Texas Advanced Computing Center (TACC) at UT Austin for providing computational resources that have contributed to the research results reported within this paper.

## Impact Statement

This paper develops a leakage free, standardized evaluation protocol for same graph cross task transfer between node classification and link prediction, and shows that transfer is often asymmetric and predictable from simple diagnostics such as homophily and baseline task learnability. The positive impact is improved rigor and reproducibility in multi objective graph learning and practical guidance for when shared encoders can reduce training and serving cost without misleading gains from protocol artifacts. Potential negative impacts include stronger inference on sensitive relational data that could enable profiling or surveillance, and the propagation or amplification of biases from node labels or observed links across tasks. We recommend careful auditing, transparent reporting of splits and negative sampling, and avoiding deployment on high stakes social graphs without governance and monitoring.

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

# A. Further Experiments and Analysis

## A.1. Hyperparameters

**Choice of $\lambda_x$ for MultiView Model.** For the Contrastive Multi-View model (Section 4.3), $\lambda_x$ is selected by validation tuning over the range $[0.2, 1.6]$, as described in Section 4 (Section 4.3). To initialize this search range, we use the graph-statistic-based formula

$$\lambda_x := 0.2 + 1.4 \left(0.55\mathbf{H_e} + 0.25\mathbf{H_n} + 0.20\mathbf{CC}\right), \tag{2}$$

clamped to $[0.2, 1.6]$. This formula provides a principled initialization that biases the search toward a reasonable region—more aggressive cross-view alignment for homophilic/clustered graphs, and more conservative coupling for heterophilic/weakly clustered graphs—but the final reported $\lambda_x$ is always the value achieving the highest validation performance, with no access to test labels. This substantially mitigates the confounding concern: the homophily-based formula shapes the search region but does not determine the final hyperparameter. Furthermore, the same directional asymmetry between NC→LP and LP→NC transfer appears consistently across WS, ET-Rep, ET-Concat, and Joint, none of which use homophily-based initialization, confirming that the pattern is not an artifact of this scheduling choice.

**Training time.** Table 6 reports wall-clock training times for each transfer regime on the Cora dataset with a GCN backbone, averaged over 10 random seeds on an Apple M3 Pro (12-core CPU, 18-core GPU, 18 GB RAM). Times are indicative of relative overhead and will vary with hardware and dataset size.

Joint's overhead is dominated by the grid search over $\lambda \in \{0.0, 0.1, 0.2, \ldots, 1.0\}$ (11 values); the per-run cost at a fixed $\lambda$ is approximately $1.80\,\text{s}$, comparable to MV. WS and ET regimes add negligible overhead beyond the source-task training cost and are omitted from the table for brevity.

*Table 6.* Wall-clock training time (seconds) per method on Cora with GCN backbone, averaged over 10 seeds (mean $\pm$ std).

| METHOD | TIME (S) |
|---|---|
| NC BASE | $0.69 \pm 0.05$ |
| LP BASE | $2.29 \pm 0.51$ |
| MV | $5.25 \pm 1.76$ |
| JOINT | $19.76 \pm 2.72$ |

**Correlation analysis across backbones.** Table 7 shows that the statistics predicting transfer are direction and backbone dependent, but with a clear overall trend for NC→LP: homophily is the most consistent predictor of LP gains. Across GCN, GraphSAGE, and GPS, correlations with node-level homophily $H_n$ are uniformly positive and frequently significant for multiple mechanisms (notably WS, and ET Concat on GCN and GPS), indicating that label-coherent neighborhoods make node-supervised representations broadly reusable for edge scoring. The effect is strongest under the GCN encoder, where representation-fusion mechanisms (ET Rep and ET Concat) exhibit the largest and most significant correlations with both $H_e$ and $H_n$, while the same mechanisms become weaker or less consistent under GraphSAGE, suggesting that backbone expressivity can reduce the direct dependence of transfer gains on homophily. In contrast, global clustering coefficient (CC) is consistently negative or near zero for NC→LP across backbones and mechanisms, implying that transitivity alone does not explain the observed LP improvements and that label alignment, rather than triangle density, is the dominant explanatory signal for NC-informed transfer.

**Combined performances per dataset.** Table 5 reports CTS for each dataset and backbone, revealing that the most effective transfer mechanism is highly dataset dependent and can vary substantially across backbones. On homophilic citation graphs, the strongest CTS values are achieved by NC-driven transfer (e.g., MV or Joint under GCN), consistent with the view that label-coherent neighborhoods make node supervision broadly useful for LP. In contrast, on heterophilic benchmarks the best CTS scores typically come from GraphSAGE or GPS with coupled objectives (MV or Joint), while ET Rep is frequently negative, highlighting the brittleness of naive representation reuse and the greater robustness of joint or contrastive coupling when homophily is low. The mixed regime further underscores the risk of negative transfer: the top CTS entries remain positive but are concentrated in specific mechanisms, while several alternatives degrade combined performance, motivating our emphasis on regime-aware choices rather than a single universal transfer recipe.

**Regime-averaged CTS trends.** Table 8 aggregates CTS by homophily regime to emphasize broad patterns that complement the per-dataset results in Table 5. On homophilic graphs, CTS is maximized by coupled training objectives, with MV and Joint achieving the highest average CTS across backbones, while ET Rep is consistently negative, reflecting that naive representation reuse can hurt one task even when the other improves. On heterophilic graphs, positive CTS concentrates in the more expressive backbones (GraphSAGE and especially GPS), where MV and Joint again provide the most reliable net

*Table 7.* **Transfer gain correlations across backbones (Pearson).** Top: Pearson correlation $r$ between *NC→LP* AUC gains (relative to each backbone's LP base model) and dataset statistics across $n = 11$ datasets. Bottom: Pearson correlation $r$ between *LP→NC* accuracy gains (relative to each backbone's NC base model) and the same statistics. Stars indicate two-sided tests for zero correlation: $^*p < 0.05$, $^{**}p < 0.01$, $^{***}p < 0.001$. $H_e$ and $H_n$ denote edge- and node-level homophily, and CC denotes global clustering coefficient.

| | **NC → LP** | | | | | | | | |
|---|---|---|---|---|---|---|---|---|---|
| | **GCN** | | | **GRAPHSAGE** | | | **GPS** | | |
| **METHOD** | $H_e$ | $H_n$ | **CC** | $H_e$ | $H_n$ | **CC** | $H_e$ | $H_n$ | **CC** |
| WS | 0.584 | **0.736**$^{**}$ | -0.336 | 0.540 | **0.689**$^{*}$ | -0.297 | 0.551 | **0.632**$^{*}$ | -0.339 |
| ET REP | **0.787**$^{**}$ | **0.847**$^{***}$ | -0.249 | 0.410 | 0.444 | -0.292 | 0.579 | **0.651**$^{*}$ | -0.289 |
| ET CONCAT | **0.838**$^{**}$ | **0.904**$^{***}$ | -0.168 | 0.474 | 0.514 | -0.345 | **0.609**$^{*}$ | **0.673**$^{*}$ | -0.250 |
| MV | 0.548 | **0.709**$^{*}$ | -0.394 | 0.440 | 0.584 | -0.432 | 0.447 | 0.573 | -0.477 |
| JOINT | 0.532 | **0.650**$^{*}$ | -0.510 | 0.435 | **0.615**$^{*}$ | -0.439 | -0.224 | -0.039 | -0.112 |

| | **LP → NC** | | | | | | | | |
|---|---|---|---|---|---|---|---|---|---|
| | **GCN** | | | **GRAPHSAGE** | | | **GPS** | | |
| **METHOD** | $H_e$ | $H_n$ | **CC** | $H_e$ | $H_n$ | **CC** | $H_e$ | $H_n$ | **CC** |
| WS | -0.275 | -0.086 | -0.548 | -0.047 | -0.155 | 0.513 | -0.220 | -0.287 | 0.052 |
| ET REP | -0.114 | -0.353 | 0.370 | 0.100 | -0.181 | **0.705**$^{*}$ | -0.015 | -0.321 | 0.518 |
| ET CONCAT | 0.270 | -0.036 | **0.782**$^{**}$ | 0.279 | 0.047 | **0.800**$^{**}$ | -0.175 | -0.463 | **0.717**$^{*}$ |
| MV | 0.279 | 0.239 | 0.380 | 0.204 | 0.173 | 0.172 | 0.444 | 0.276 | 0.311 |
| JOINT | -0.059 | -0.241 | 0.428 | 0.235 | -0.008 | **0.780**$^{**}$ | -0.188 | -0.380 | 0.119 |

*Table 8.* **Combined performance.** CoTask Score (CTS) summarizes joint NC+LP performance as the average percent improvement over a fixed GCN reference across the two tasks. We group datasets by homophily regime (homophilic, heterophilic, mixed) and report the mean CTS within each group. The full per-dataset CTS results and group membership are in Table 5. For each row, we highlight the **best**, **second**, and **third** CTS values.

| | GCN backbone | | | | | | GraphSAGE backbone | | | | | | GPS backbone | | | | | |
|---|---|---|---|---|---|---|---|---|---|---|---|---|---|---|---|---|---|---|
| Regime | Base | WS | ET Rep | ET Con | MV | Joint | Base | WS | ET Rep | ET Con | MV | Joint | Base | WS | ET Rep | ET Con | MV | Joint |
| Hom | 0.0 | 5.8 | -4.9 | 5.2 | **7.1** | **7.3** | -1.4 | 3.7 | -11.4 | 2.7 | 5.0 | 5.1 | -2.4 | -0.8 | -11.7 | -0.9 | **6.0** | **6.0** |
| Het | 0.0 | 2.0 | -9.3 | -1.9 | 0.2 | 3.6 | 15.7 | **24.3** | 3.4 | 19.2 | **24.7** | **23.3** | 8.2 | 11.5 | -0.6 | 13.3 | 13.5 | **27.7** |
| Mixed | 0.0 | 0.1 | **7.6** | **5.3** | -1.1 | **6.9** | -14.1 | -6.7 | 2.9 | **3.9** | -9.0 | 1.6 | -21.0 | -13.3 | 2.5 | 2.7 | -10.1 | -2.7 |

gains, suggesting that coupling objectives is a robust hedge against negative transfer when homophily is low. Finally, the mixed regime highlights that aggregated performance can be dominated by dataset-specific effects: averages are positive for GCN via ET Rep and ET Concat, but several backbone-mechanism combinations remain negative, reinforcing that neither transfer direction nor mechanism is universally beneficial and motivating regime-aware selection.

### A.2. Additional Detail and Definitions

*Edge homophily* $H_e$ is the fraction of edges that connect nodes sharing the same label,

$$H_e := \frac{|\{(u,v) \in E : y_u = y_v\}|}{|E|},$$

and *node homophily* $H_n$ is the average over nodes of the fraction of same-label neighbors,

$$H_n := \frac{1}{|V|} \sum_{v \in V} \frac{|\{u \in \mathcal{N}(v) : y_u = y_v\}|}{|\mathcal{N}(v)|},$$

where $\mathcal{N}(v)$ denotes the neighbors of $v$.

### A.3. Sanity checks for evaluation protocol

We verify the following for every dataset and run:

- **Fixed splits:** node splits $(V_{\text{tr}}^{\text{NC}}, V_{\text{va}}^{\text{NC}}, V_{\text{te}}^{\text{NC}})$ and LP positive edge splits $(E_{\text{tr}}^+, E_{\text{va}}^+, E_{\text{te}}^+)$ are generated once and reused.

*Table 9.* **NC→LP inductive transfer results (AUC).** We report LP transfer gains (in percentage points) from NC-driven transfer. **Base** is the single-task LP AUC; other entries are gains relative to that base. For each dataset and backbone, we bold the largest gain. MV and Joint are reported for GCN only.

| | | | | GCN | | | | | | GraphSAGE | | | | GPS | | |
|---|---|---|---|---|---|---|---|---|---|---|---|---|---|---|---|---|
| **Dataset** | $H_e$ | $H_n$ | **CC** | **Base** | **WS** | **ET Rep** | **ET Con** | **MV** | **Joint** | **Base** | **WS** | **ET Rep** | **ET Con** | **Base** | **WS** | **ET Rep** | **ET Con** |
| Cora | 0.81 | 0.83 | 0.09 | 63.89 | 16.00 | 17.16 | 15.98 | 20.31 | **21.37** | 55.89 | **13.35** | 7.23 | 8.37 | 63.90 | 9.88 | **16.33** | 13.75 |
| Citeseer | 0.74 | 0.71 | 0.13 | 66.57 | 17.20 | 15.07 | 16.46 | **22.26** | 19.79 | 53.66 | **16.52** | 8.58 | 9.09 | 66.44 | 9.70 | **14.61** | 11.01 |
| PubMed | 0.80 | 0.79 | 0.05 | 73.07 | **14.94** | 11.92 | 12.59 | 16.51 | 16.34 | 64.93 | -1.39 | -4.31 | -2.90 | 71.02 | **10.89** | 9.99 | 10.62 |
| Texas | 0.11 | 0.07 | 0.03 | 62.29 | **2.23** | -2.55 | -1.02 | 4.86 | -12.58 | 63.36 | 1.09 | 1.76 | **3.54** | 66.22 | **1.38** | -2.47 | 1.28 |
| Cornell | 0.13 | 0.11 | 0.03 | 58.51 | **8.39** | 0.70 | 2.13 | 7.14 | 6.91 | 56.22 | 2.81 | 0.90 | **3.03** | 59.86 | -0.73 | -1.14 | **0.25** |
| Wisconsin | 0.20 | 0.17 | 0.04 | 64.26 | 2.19 | -6.48 | -3.25 | 2.40 | 2.40 | 63.91 | 0.65 | 2.18 | **2.78** | 64.09 | 1.59 | 0.32 | **4.73** |
| Actor | 0.22 | 0.22 | 0.02 | 62.19 | -0.02 | -2.80 | -4.29 | **1.37** | 2.24 | 62.21 | 0.41 | **3.10** | 1.95 | 63.50 | **0.66** | -1.57 | -0.48 |
| roman-empire | 0.05 | 0.05 | 0.29 | 56.29 | **-0.48** | -5.32 | -5.18 | -3.88 | -4.06 | 49.19 | **0.71** | -0.82 | -2.71 | 53.71 | **-0.93** | -2.63 | -1.54 |
| USA | 0.70 | 0.37 | 0.43 | 80.47 | **-0.04** | -4.27 | -4.49 | 0.92 | 2.60 | 77.76 | **1.16** | -4.94 | -2.24 | 81.22 | **-1.44** | -4.05 | -3.32 |
| Europe | 0.45 | 0.27 | 0.33 | 78.48 | **-1.78** | -13.62 | -13.78 | -0.60 | 0.73 | 80.01 | **-1.33** | -4.37 | -2.56 | 78.05 | **-1.22** | -3.45 | -2.02 |
| Brazil | 0.40 | 0.22 | 0.45 | 75.00 | **-0.80** | -14.54 | -13.83 | -1.33 | 1.05 | 75.89 | **-1.16** | -3.40 | -2.72 | 73.81 | **-3.22** | -6.16 | -6.26 |

- **No edge leakage:** message passing uses only $E_{\mathrm{tr}}^+$ (i.e., $\mathrm{Edges}(A_{\mathrm{obs}}) \subseteq E_{\mathrm{tr}}^+$), and evaluation positives satisfy $(E_{\mathrm{va}}^+ \cup E_{\mathrm{te}}^+) \cap \mathrm{Edges}(A_{\mathrm{obs}}) = \emptyset$.

- **Valid fixed negatives:** for each split $s \in \{\mathrm{tr}, \mathrm{va}, \mathrm{te}\}$, negatives satisfy $E_s^- \cap E = \emptyset$ and are reused across methods and runs.

- **Repeatability:** rerunning with the same seed reproduces identical splits and negatives.

### A.4. Inductive setting

**Setup.** We partition nodes into disjoint sets $V = V_{\mathrm{tr}} \dot\cup V_{\mathrm{va}} \dot\cup V_{\mathrm{te}}$, where $V_{\mathrm{te}}$ is unseen during training. For NC, we set $V_{\mathrm{tr}}^{\mathrm{NC}} := V_{\mathrm{tr}}$, $V_{\mathrm{va}}^{\mathrm{NC}} := V_{\mathrm{va}}$, and $V_{\mathrm{te}}^{\mathrm{NC}} := V_{\mathrm{te}}$. Training uses only features and supervision on $V_{\mathrm{tr}}$; validation and test are performed on $V_{\mathrm{va}}$ and $V_{\mathrm{te}}$.

**Training subgraph and LP positives.** We define the training edge set as the induced subgraph on $V_{\mathrm{tr}}$:

$$E_{\mathrm{tr}} := \{(u, v) \in E : \ u, v \in V_{\mathrm{tr}}\}.$$

We split $E_{\mathrm{tr}}$ into LP positives $E_{\mathrm{tr}}^+$ and $E_{\mathrm{va}}^+$. Training message passing uses only $E_{\mathrm{tr}}^+$: $A_{\mathrm{tr}} := \mathrm{Adj}(V_{\mathrm{tr}}, E_{\mathrm{tr}}^+)$.

**Test edges: observed context versus evaluated positives.** Let $E_{\mathrm{te}}^{\mathrm{all}} := \{(u, v) \in E : \ u \in V_{\mathrm{te}} \text{ or } v \in V_{\mathrm{te}}\}$ denote edges incident to test nodes. To allow test nodes to aggregate from observed neighborhoods without leaking the specific positives being evaluated, we split $E_{\mathrm{te}}^{\mathrm{all}}$ into two disjoint sets $E_{\mathrm{te}}^{\mathrm{all}} = E_{\mathrm{te}}^{\mathrm{obs}} \dot\cup E_{\mathrm{te}}^{\mathrm{pred}}$, where $E_{\mathrm{te}}^{\mathrm{pred}}$ are the evaluated positive edges for inductive LP, and $E_{\mathrm{te}}^{\mathrm{obs}}$ are additional observed edges that may be used for message passing at evaluation time. We construct this split once with a fixed seed using a fixed fraction policy (reported with experiments) and reuse it for every method.

**Results in Inductive Setting.** Tables 9 and 10 summarize same-graph cross-task transfer in both directions using an *inductive* setting. We report results for all five transfer regimes under GCN, and for WS, ET-Rep, and ET-Concat under GraphSAGE and GPS. The results echo the same conclusions drawn in Section 5. NC→LP remains a strong transfer direction in homophilic contexts across all mechanisms: on Cora, Citeseer, and PubMed, MV and Joint achieve the largest GCN gains (e.g., MV/Joint of 22.26/19.79 on Citeseer and 20.31/21.37 on Cora), consistent with the transductive pattern where coupled training outperforms post-hoc reuse. Performance is close to baseline in heterophilic or mixed environments. LP→NC remains much weaker inductively: gains from MV and Joint on homophilic datasets are small and inconsistent (e.g., 1.56/−0.40 on Cora, 1.90/−0.92 on Citeseer, −0.47/−0.11 on PubMed), confirming that the fragility of LP→NC transfer is not a transductive artifact. Overall, performance across all datasets is lower in the inductive setting than the transductive setting, reflecting the harder generalization requirement. Extending MV and Joint to GraphSAGE and GPS backbones in the inductive setting remains future work.

### A.5. Intuition: shared latent space vs. shared parameter space

**Shared latent space (representation viewpoint).** Most GNNs and graph transformers can be decomposed into an encoder and a task-specific head. Given features $X$ and a protocol adjacency $A$, an encoder $E_\theta$ produces node embeddings

*Table 10.* **LP→NC inductive transfer results (Acc).** We report NC transfer gains (in percentage points) from LP-driven transfer. **Base** is the single-task NC accuracy; other entries are gains relative to that base. For each dataset and backbone, we bold the largest gain. MV and Joint are reported for GCN only.

| | | | | GCN | | | | | GraphSAGE | | | | GPS | | |
|---|---|---|---|---|---|---|---|---|---|---|---|---|---|---|---|
| **Dataset** | **H$_e$** | **H$_n$** | **CC** | **Base** | **WS** | **ET Rep** | **ET Con** | **MV** | **Joint** | **Base** | **WS** | **ET Rep** | **ET Con** | **Base** | **WS** | **ET Rep** | **ET Con** |
| Cora | 0.81 | 0.83 | 0.09 | 71.66 | **0.88** | -28.44 | 0.60 | 1.56 | -0.40 | 70.59 | 0.48 | -37.86 | **0.94** | 67.55 | -1.86 | -37.88 | **0.65** |
| Citeseer | 0.74 | 0.71 | 0.13 | 68.18 | **0.62** | -24.91 | -0.27 | 1.90 | -0.92 | 68.21 | **1.08** | -36.23 | 0.62 | 63.49 | -5.29 | -38.26 | **-1.17** |
| PubMed | 0.80 | 0.79 | 0.05 | 86.53 | **0.12** | -27.54 | -0.10 | -0.47 | -0.11 | 87.41 | **-0.06** | -33.94 | -1.03 | 85.11 | **-2.80** | -39.09 | -12.01 |
| Texas | 0.11 | 0.07 | 0.03 | 82.63 | -1.05 | -15.00 | -2.89 | 0.53 | **2.11** | 80.26 | **-0.52** | -15.79 | -3.68 | 48.42 | 3.42 | -7.63 | **16.84** |
| Cornell | 0.13 | 0.11 | 0.03 | 78.16 | **4.21** | -13.95 | -1.58 | 2.10 | -0.79 | 75.79 | **1.32** | -22.37 | -0.26 | 48.68 | 2.90 | -12.63 | **11.85** |
| Wisconsin | 0.20 | 0.17 | 0.04 | 81.76 | -1.77 | -18.04 | -2.75 | 0.40 | **-0.20** | 80.59 | **1.57** | -16.47 | -3.53 | 57.65 | 0.39 | -15.69 | **6.66** |
| Actor | 0.22 | 0.22 | 0.02 | 33.79 | **0.05** | -2.29 | -0.42 | 0.12 | 0.43 | 35.38 | **0.42** | -6.08 | -0.12 | 32.53 | **-0.19** | -4.11 | -2.17 |
| roman-empire | 0.05 | 0.05 | 0.29 | 63.71 | **-0.06** | -18.28 | -0.36 | -1.00 | -0.25 | 65.48 | **-0.18** | -25.99 | -1.54 | 61.73 | **-0.31** | -23.83 | -10.71 |
| USA | 0.70 | 0.37 | 0.43 | 23.32 | 1.13 | 1.22 | 1.22 | 1.01 | **1.85** | 24.33 | 0.04 | 0.04 | **1.13** | 24.83 | **2.14** | -0.17 | 0.34 |
| Europe | 0.45 | 0.27 | 0.33 | 24.94 | **0.00** | -0.99 | **0.00** | 0.00 | -0.13 | 24.57 | -0.50 | **0.49** | -0.13 | 25.93 | **-1.61** | -1.98 | -1.86 |
| Brazil | 0.40 | 0.22 | 0.45 | 23.70 | 0.37 | **1.86** | 1.86 | -0.37 | **3.71** | 27.04 | **0.37** | -3.34 | -5.19 | 25.93 | **-0.37** | -1.49 | -1.49 |

$Z_\theta = E_\theta(X, A) \in \mathbb{R}^{|V| \times d}$. Downstream tasks differ mainly in how they *read out* or *score* these embeddings: node classification applies a node-wise classifier to $z_v$, link prediction scores pairs $(z_u, z_v)$ via an edge decoder, and graph classification pools $\{z_v\}_{v \in V}$ into a graph-level vector. This makes cross-task transfer plausible: supervision from one task shapes $Z_\theta$ in ways that can either help the other task (when the induced geometry aligns) or hurt it (negative transfer when the objectives prefer incompatible geometries).

**Shared parameter space (optimization viewpoint).** A complementary perspective, illustrated in Figure 1, is that NC and LP are optimized over the *same parameter space* for a fixed backbone architecture. Fix an encoder family $\{E_\theta : \theta \in \Theta\}$ and consider two training objectives under our leakage-free protocol: $\mathcal{L}_{\text{NC}}(\theta)$ (node supervision) and $\mathcal{L}_{\text{LP}}(\theta)$ (edge supervision with fixed negatives). Training on NC seeks parameters $\theta_{\text{NC}} \in \arg\min_{\theta \in \Theta} \mathcal{L}_{\text{NC}}(\theta)$, while training on LP seeks $\theta_{\text{LP}} \in \arg\min_{\theta \in \Theta} \mathcal{L}_{\text{LP}}(\theta)$. Because these losses can have different local minima and basins of attraction, transfer mechanisms can be viewed as different ways of moving through $\Theta$: warm start initializes the target optimization near a source minimizer, embedding transfer injects source representations while re-optimizing the target, and MV or Joint explicitly couple objectives to bias optimization toward regions where the induced representations are simultaneously useful.

**How this relates to our findings.** This parameter-space picture explains why transfer can be directional and regime dependent. If NC and LP prefer compatible regions of $\Theta$ on a given dataset regime, initialization or coupling can yield positive transfer; if they prefer incompatible regions, transfer can be fragile and negative, especially for post hoc reuse. We emphasize that this discussion is an *intuition* for optimization and representation compatibility, not a theoretical guarantee; all claims in the paper are supported by the leakage-free protocol and empirical results.

**Representational geometry evidence.** To move beyond intuition, we provide quantitative evidence for why NC→LP transfer gains are larger on homophilic graphs. For each dataset, we compute the mean cosine similarity gap between linked and unlinked node pairs using (i) raw input features and (ii) NC-trained embeddings (GCN backbone, ET-Rep). Formally, for a set of positive pairs $\mathcal{P}$ and negative pairs $\mathcal{N}$, the gap is

$$\Delta := \frac{1}{|\mathcal{P}|} \sum_{(u,v) \in \mathcal{P}} \cos(z_u, z_v) - \frac{1}{|\mathcal{N}|} \sum_{(u,v) \in \mathcal{N}} \cos(z_u, z_v),$$

where $z_u, z_v$ are either raw feature vectors or NC-learned embeddings. Table 11 reports the gap under raw features (**Feat Gap**) and NC-learned embeddings (**Emb Gap**), together with the absolute increase.

NC training amplifies the cosine similarity gap across all datasets, but the resulting embedding gap is substantially larger on homophilic graphs (mean increase $0.61$) than on heterophilic ones (mean increase $0.29$), a difference of approximately 2.1 times. This provides direct geometric evidence for the NC→LP asymmetry: on homophilic graphs, NC supervision shapes an embedding space where linked pairs are far more separable from unlinked pairs, making the learned representations naturally compatible with LP decoding. On heterophilic graphs, NC training still improves separability, but the resulting geometry is less aligned with edge formation patterns, explaining the weaker and less consistent NC→LP gains in those regimes.

*Table 11.* Cosine similarity gap between linked and unlinked pairs under raw features vs. NC-trained embeddings (GCN, ET-Rep). A larger gap indicates that linked pairs are more separable, making embeddings more compatible with LP decoding.

| DATASET | $H_n$ | FEAT GAP | EMB GAP | INCREASE |
|---------|-------|----------|---------|----------|
| CITESEER | 0.706 | 0.1475 | 0.8932 | +0.7457 |
| PUBMED | 0.792 | 0.1991 | 0.6902 | +0.4911 |
| CORA | 0.825 | 0.1120 | 0.7876 | +0.6756 |
| TEXAS | 0.057 | 0.0155 | 0.2681 | +0.2526 |
| CORNELL | 0.111 | 0.0272 | 0.3218 | +0.2946 |
| WISCONSIN | 0.155 | 0.0331 | 0.2793 | +0.2462 |
| ACTOR | 0.220 | $-0.0081$ | 0.2920 | +0.3001 |

## A.6. Hyperparameter Sensitivity

The main results in Section 5 use a fixed hyperparameter configuration (2-layer GNN, hidden dimension 64, dropout 0.5, learning rate 0.01, early stopping patience 50) tuned once on validation and shared across all transfer regimes. A natural concern is whether the directional asymmetry and regime-specific patterns we report are artifacts of this particular configuration. To assess sensitivity, we re-ran the full transductive NC↔LP transfer study under an alternative configuration: 3 layers, hidden dimension 32, dropout 0.3, learning rate 0.01. Tables 12 and 13 report results under this alternative setting using a GCN backbone.

The key patterns are preserved. For NC→LP (Table 12), gains remain consistently positive on homophilic graphs (CORA, CITESEER, PUBMED) across all five transfer mechanisms, with Joint and ET Rep achieving the largest improvements. On heterophilic datasets, gains are smaller and more variable, with ET Rep and ET Concat occasionally negative, mirroring the main results. For LP→NC (Table 13), the fragility observed in Section 5 persists: ET Rep produces severe accuracy drops on homophilic datasets (e.g., $-33.2$ pp on CORA, $-39.8$ pp on PUBMED), while other mechanisms produce near-zero or modestly negative gains. The structure-dominant datasets (USA, BRAZIL) again show the most favorable LP→NC outcomes.

Quantitatively, the magnitude of some gains shifts modestly between configurations—reflecting the expected sensitivity of absolute performance to depth and width—but the directional asymmetry between NC→LP and LP→NC, the ranking of mechanisms within each direction, and the regime-dependent pattern are stable. This consistency across two architecturally distinct configurations supports the conclusion that the observed transfer behavior reflects properties of the dataset regimes and task interaction, rather than a particular hyperparameter choice.

*Table 12.* NC→LP transfer results (AUC). We report the LP base AUC (mean $\pm$ std) and transfer gains (in percentage points, $\pm$ std of the method) relative to the LP single-task base model. For each dataset, we **bold** the largest gain.

| DATASET | $H_e$ | $H_n$ | BASE | WS | ET REP | ET CON | MV | JOINT |
|---------|-------|-------|------|----|--------|--------|----|----|
| CORA | 0.81 | 0.83 | $75.9 \pm 2.4$ | $+11.7 \pm 1.1$ | $+14.9 \pm 0.8$ | $+10.9 \pm 1.9$ | $+12.9 \pm 1.6$ | $\mathbf{+15.6 \pm 0.9}$ |
| CITESEER | 0.74 | 0.71 | $74.3 \pm 2.0$ | $+11.6 \pm 1.5$ | $+13.8 \pm 1.0$ | $+9.5 \pm 2.4$ | $+10.5 \pm 1.4$ | $\mathbf{+14.9 \pm 1.2}$ |
| PUBMED | 0.80 | 0.79 | $89.1 \pm 0.4$ | $+5.5 \pm 0.5$ | $+3.8 \pm 0.5$ | $+3.5 \pm 0.8$ | $+1.4 \pm 2.2$ | $\mathbf{+6.1 \pm 0.3}$ |
| TEXAS | 0.11 | 0.07 | $65.9 \pm 6.6$ | $+1.4 \pm 8.4$ | $+2.8 \pm 7.2$ | $-4.3 \pm 7.4$ | $+0.2 \pm 6.2$ | $\mathbf{+4.8 \pm 5.5}$ |
| CORNELL | 0.13 | 0.11 | $73.0 \pm 8.7$ | $\mathbf{+5.5 \pm 8.1}$ | $-1.5 \pm 6.0$ | $-5.9 \pm 8.8$ | $-2.0 \pm 3.6$ | $+2.3 \pm 9.4$ |
| WISCONSIN | 0.20 | 0.17 | $71.2 \pm 7.7$ | $+4.5 \pm 5.8$ | $+2.5 \pm 5.6$ | $+2.8 \pm 5.3$ | $+6.7 \pm 6.8$ | $\mathbf{+7.6 \pm 5.6}$ |
| ACTOR | 0.22 | 0.22 | $80.4 \pm 0.6$ | $\mathbf{+1.2 \pm 2.0}$ | $-0.9 \pm 1.0$ | $-0.5 \pm 0.8$ | $+0.1 \pm 0.7$ | $+0.2 \pm 0.8$ |
| ROMAN | 0.05 | 0.05 | $73.3 \pm 1.2$ | $\mathbf{+3.2 \pm 0.7}$ | $-7.3 \pm 1.6$ | $-6.1 \pm 1.6$ | $-12.8 \pm 2.0$ | $-10.6 \pm 0.9$ |
| USA | 0.70 | 0.37 | $94.7 \pm 1.5$ | $\mathbf{+0.8 \pm 0.4}$ | $\mathbf{+0.8 \pm 0.5}$ | $\mathbf{+0.8 \pm 0.5}$ | $-2.0 \pm 1.4$ | $+0.2 \pm 0.8$ |
| EUROPE | 0.45 | 0.27 | $92.2 \pm 0.7$ | $\mathbf{+0.2 \pm 0.8}$ | $-0.5 \pm 0.8$ | $-0.1 \pm 0.8$ | $-1.9 \pm 1.0$ | $-2.1 \pm 1.4$ |
| BRAZIL | 0.41 | 0.22 | $89.8 \pm 1.8$ | $+0.8 \pm 2.1$ | $+0.6 \pm 2.0$ | $\mathbf{+1.0 \pm 1.6}$ | $-0.7 \pm 1.6$ | $+0.1 \pm 1.3$ |

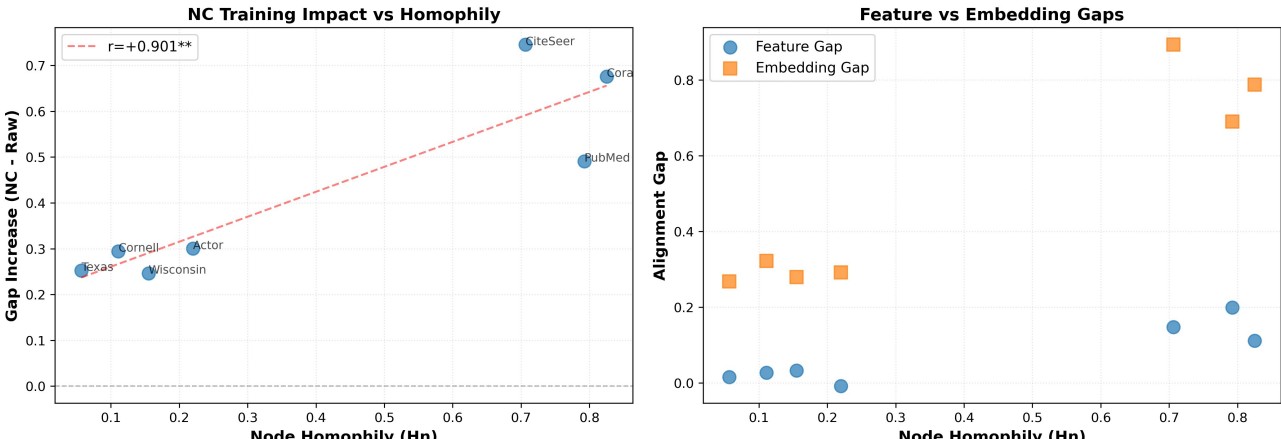

*Figure 2.* **NC training amplifies cosine similarity gaps on homophilic graphs.** *Left:* The increase in cosine similarity gap between linked and unlinked pairs after NC training (versus raw features) correlates strongly with node homophily ($r = +0.901^{**}$), explaining why NC→LP transfer is reliably beneficial on homophilic datasets. *Right:* On homophilic graphs, NC-trained embeddings (orange) produce substantially larger alignment gaps than raw features (blue), whereas on heterophilic graphs the two are comparable. This geometric evidence supports the mechanism described in Section 5: NC supervision shapes an embedding space where linked pairs are more separable from unlinked pairs, and this effect is strongest precisely where NC→LP transfer succeeds.

### A.7. Additional Homophilic Datasets: Amazon-Photo and Amazon-Computers

To further validate that NC→LP transfer generalizes beyond citation graphs, we evaluated the same leakage-free protocol on two additional homophilic co-purchase graphs from the Amazon benchmark suite: PHOTO ($H_e = 0.83$, $H_n = 0.85$) and COMPUTERS ($H_e = 0.78$, $H_n = 0.80$). These graphs differ from the citation networks in Table 1 in their construction (product co-purchase rather than citation links) and density, providing a complementary test of the homophily-driven NC→LP transfer hypothesis.

Table 14 reports bidirectional transfer results under the standard GCN configuration (2 layers, hidden dimension 64, dropout 0.5, learning rate 0.01, early stopping patience 50).

Results are consistent with the main findings. NC→LP gains are positive across all mechanisms on both datasets, with ET Rep achieving the largest improvement on PHOTO (+3.2 pp AUC) and COMPUTERS (+3.8 pp AUC), in line with the strong ET Rep performance on other homophilic graphs. The absolute magnitude of gains is somewhat smaller than on the citation graphs, which we attribute to the already-high LP baseline AUC (94.6 and 93.7 respectively), leaving less headroom for improvement. Importantly, LP→NC transfers are largely neutral or slightly negative: ET Rep again degrades accuracy ($-4.3$ pp on PHOTO, $-14.3$ pp on COMPUTERS), while other mechanisms produce near-zero gains. This replicates the asymmetry seen on CORA, CITESEER, and PUBMED, and reinforces that NC→LP is the reliable transfer direction on homophilic graphs regardless of the specific graph domain, while LP→NC remains fragile even when homophily is high.

*Table 13.* LP→NC transfer results (Accuracy). We report the NC base accuracy (mean ± std) and transfer gains (in percentage points, ± std of the method) relative to the NC single-task base model. For each dataset, we **bold** the largest gain.

| DATASET | $H_e$ | $H_n$ | BASE | WS | ET REP | ET CON | MV | JOINT |
|---|---|---|---|---|---|---|---|---|
| CORA | 0.81 | 0.83 | $84.5 \pm 1.4$ | $\mathbf{+0.0 \pm 1.6}$ | $-33.2 \pm 6.0$ | $+0.3 \pm 2.1$ | $-0.9 \pm 1.2$ | $-0.5 \pm 1.6$ |
| CITESEER | 0.74 | 0.71 | $71.5 \pm 1.9$ | $-0.4 \pm 1.4$ | $-34.1 \pm 3.3$ | $-0.3 \pm 2.4$ | $-2.4 \pm 2.2$ | $-0.4 \pm 1.8$ |
| PUBMED | 0.80 | 0.79 | $87.2 \pm 0.6$ | $\mathbf{+0.0 \pm 0.7}$ | $-39.8 \pm 7.2$ | $-0.3 \pm 0.8$ | $-0.5 \pm 0.5$ | $-0.4 \pm 0.4$ |
| TEXAS | 0.11 | 0.07 | $47.4 \pm 8.0$ | $+2.9 \pm 6.3$ | $\mathbf{+12.1 \pm 6.2}$ | $+1.8 \pm 8.5$ | $+1.6 \pm 7.9$ | $+4.2 \pm 8.3$ |
| CORNELL | 0.13 | 0.11 | $41.3 \pm 7.3$ | $+0.0 \pm 6.3$ | $+3.1 \pm 5.3$ | $-1.1 \pm 6.2$ | $\mathbf{+1.3 \pm 6.0}$ | $-0.8 \pm 6.5$ |
| WISCONSIN | 0.20 | 0.17 | $43.3 \pm 4.8$ | $+2.0 \pm 6.1$ | $\mathbf{+6.5 \pm 5.8}$ | $-0.2 \pm 4.5$ | $-2.0 \pm 4.9$ | $+4.7 \pm 4.1$ |
| ACTOR | 0.22 | 0.22 | $26.9 \pm 1.6$ | $+0.9 \pm 1.3$ | $-1.4 \pm 1.1$ | $+0.3 \pm 1.7$ | $-0.3 \pm 1.3$ | $\mathbf{+1.1 \pm 1.2}$ |
| ROMAN | 0.05 | 0.05 | $39.5 \pm 0.9$ | $-1.1 \pm 1.3$ | $-20.1 \pm 0.9$ | $-0.2 \pm 0.7$ | $-4.6 \pm 3.2$ | $\mathbf{+0.1 \pm 0.4}$ |
| USA | 0.70 | 0.37 | $55.6 \pm 3.6$ | $-0.1 \pm 3.6$ | $+5.5 \pm 3.3$ | $\mathbf{+6.5 \pm 4.1}$ | $-1.7 \pm 3.1$ | $+3.1 \pm 3.2$ |
| EUROPE | 0.45 | 0.27 | $55.1 \pm 2.9$ | $-0.4 \pm 3.1$ | $-0.6 \pm 5.9$ | $-0.5 \pm 3.7$ | $-1.6 \pm 6.5$ | $\mathbf{+1.2 \pm 6.8}$ |
| BRAZIL | 0.41 | 0.22 | $55.2 \pm 7.3$ | $+4.8 \pm 6.9$ | $+10.7 \pm 6.0$ | $\mathbf{+11.1 \pm 8.1}$ | $-1.5 \pm 10.9$ | $+10.0 \pm 9.4$ |

*Table 14.* For each dataset, *top row*: NC→LP gains in AUC (pp) relative to the LP single-task base; *bottom row*: LP→NC gains in Accuracy (pp) relative to the NC single-task base. **Bold** indicates the largest gain per row.

| DATASET | TASK | $H_e$ | $H_n$ | BASE | WS | ET REP | ET CON | MV | JOINT |
|---|---|---|---|---|---|---|---|---|---|
| PHOTO | NC→LP (AUC) | 0.83 | 0.85 | $94.6 \pm 1.0$ | $+2.5 \pm 0.8$ | $\mathbf{+3.2 \pm 0.1}$ | $+2.7 \pm 0.3$ | $+1.6 \pm 0.2$ | $+2.8 \pm 0.1$ |
|  | LP→NC (ACC) |  |  | $93.0 \pm 1.6$ | $+0.8 \pm 0.7$ | $-4.3 \pm 0.9$ | $+0.6 \pm 0.7$ | $-0.1 \pm 0.8$ | $\mathbf{+1.0 \pm 0.7}$ |
| COMPUTERS | NC→LP (AUC) | 0.78 | 0.80 | $93.7 \pm 0.8$ | $+0.2 \pm 1.0$ | $\mathbf{+3.8 \pm 0.1}$ | $+2.2 \pm 0.3$ | $+1.4 \pm 0.6$ | $+2.5 \pm 0.5$ |
|  | LP→NC (ACC) |  |  | $89.1 \pm 0.7$ | $\mathbf{+0.0 \pm 0.3}$ | $-14.3 \pm 2.3$ | $-0.6 \pm 1.1$ | $-1.3 \pm 0.6$ | $-0.2 \pm 1.1$ |

