# OpenReview forum: "Same Graph Cross-Task Transfer in GNNs: Protocols and Predictors"
_ICML.cc/2026/Conference — ICML 2026 regular_

### Official Review · Reviewer_dY7S · 2026-03-09

**Soundness:** 2
**Presentation:** 3
**Significance:** 2
**Originality:** 3
**Overall Recommendation:** 4
**Confidence:** 3

**Summary:**

This paper addresses the issue of unreliable conclusions in cross-task transfer within Graph Neural Networks (GNNs) when handling node classification (NC) and link prediction (LP) tasks on the same graph structure, often caused by inconsistent data splits, loose evaluation protocols, and potential "label leakage." The authors propose a standardized "Leakage-free Protocol" that ensures fairness in cross-task evaluation by fixing node/edge splits and strictly isolating test edges from the message-passing graph. Additionally, the study evaluates several lightweight transfer mechanisms (such as warm-start initialization, embedding transfer, and joint training) and introduces a comprehensive evaluation metric called CTS. The results demonstrate that cross-task transfer is highly directional and scenario-dependent: in homophilic graphs, leveraging node labels to enhance link prediction (NC → LP) is consistently effective; conversely, using link prediction to improve node classification (LP → NC) is only effective in structure-driven scenarios where the prediction tasks are not yet saturated.

**Compliance With Llm Reviewing Policy:**

Affirmed.

**Final Justification:**

The authors have addressed most of my concerns, so I will keep my positive score.

**Key Questions For Authors:**

1 The leakage-free protocol enforces fixed NC and LP splits, sacrificing the inherent partition settings of each task. In real-world research or industrial deployment where tasks use their respective "optimal splits," do the transfer patterns discovered under this protocol still hold true?

2The paper’s core conclusions are based on a fully supervised NC setting with 60% training nodes. However, the heart of GNN node classification research lies in few-shot semi-supervised settings. Will the model selection guidelines proposed in this paper remain effective when supervision is extremely sparse?

3The paper suggests that homophily can guide the choice of transfer direction. Is it feasible to develop an automated framework that detects graph features during the early stages of training and dynamically adjusts the weights ($\lambda$) for NC and LP tasks accordingly?

4 Can this leakage-free protocol be extended to other graph tasks, such as subgraph classification or graph-level prediction? When combining different types of tasks, how should a unified "leakage-free" boundary be defined?

**Limitations:**

yes

**Strengths And Weaknesses:**

Strengths

1 To address the frequently overlooked "data leakage" issue in graph learning (e.g., test edges participating in feature aggregation), the paper proposes a strict isolation protocol, establishing a reliable foundation for cross-task research.

2The experiments cover three mainstream backbone networks (GCN, GraphSAGE, GPS) and a diverse range of datasets (homophilic, heterophilic, and structure-driven), ensuring the conclusions possess strong generalizability.

3 By utilizing simple data statistical indicators such as homophily, the paper predicts when and in which direction transfer should occur, providing a solid basis for selecting model architectures in real-world applications.

Weaknesses
1 The paper is primarily an empirical study. While it provides intuitive explanations for its findings, it lacks rigorous mathematical proofs or generalizable error bound analysis. It does not formally quantify how much supervision signal from NC can compensate for structural information in LP from an information-theoretic perspective.

2 It overlooks other critical graph tasks such as Graph Classification, Graph Clustering, or Graph Regression. It also fails to explore cross-hierarchical transfer, such as how local node-level supervision influences global graph-level properties.

---

> ### Author Rebuttal · Authors · 2026-03-28
>
> **Thank you for the insightful feedback. We address the main concerns below.**
>
> ### W1. No theoretical discussions
>
> We respectfully disagree that the paper lacks rigor because it does not include a formal theory. The contribution of this work is to formalize same-graph NC↔LP transfer, remove protocol confounds that invalidate prior comparisons, and **establish a robust directional asymmetry under controlled evaluation**. The explanation is also not merely intuitive: the paper provides two complementary mechanism-level views, *shared latent space* and *shared parameter space*, which explain why NC→LP is favored on homophilic graphs while LP→NC is fragile except when LP is already highly learnable from structure and NC remains unsaturated. These claims are supported by **systematic evidence across multiple backbones, transfer mechanisms, graph regimes**, and both transductive and inductive settings. An information-theoretic or generalization-bound treatment would be valuable future work, but its absence does not make the current contribution insufficiently rigorous.
>
> ---
>
> ### W2/Q4. Other graph tasks and protocol extension
>
> We view this as **a question of scope rather than a weakness**. Our claim is not that all graph task pairs exhibit the same asymmetry — it is specific to the same-graph setting where two supervision sources naturally coexist on one fixed graph, and NC and LP are the most natural and practically important such pair. Standard graph classification and graph regression are not same-graph tasks, since they are defined over collections of graphs rather than multiple objectives on one fixed graph. Tasks such as community detection or subgraph-level prediction are closer in spirit, but they introduce **substantially harder protocol questions** around overlap, leakage, and compatible splits that are not minor extensions of our setup and likely require separate paper-length treatment.
>
> The core leakage-free principle is nonetheless general: **evaluation structures should not appear in the graph used to compute representations**. For NC and LP, this means excluding validation and test edges from the observed message-passing graph. In principle the same idea extends to other tasks, but the definition of an evaluation structure becomes task-dependent. For subgraph-level tasks, overlap among subgraphs creates new coordination and leakage challenges. For graph-level prediction, the setting is fundamentally different because supervision is defined over multiple independent graphs. We will position these as natural but non-trivial extensions in the revision.
>
> ---
>
> ### Q1. Adapting to real-world splits
>
> The fixed NC and LP splits are intentional because they **isolate task compatibility from supervision-budget effects**: if the two tasks use different data budgets or observed graphs, apparent transfer gains can reflect data availability rather than genuine cross-task mechanisms. We do not claim that exact transfer magnitudes will remain identical under arbitrary task-specific splits. Rather, our claim is that the qualitative asymmetry is driven by underlying task compatibility rather than split-specific artifacts. NC→LP gains on homophilic graphs arise from label-clustered geometry learned by NC, while LP→NC remains much less reliable unless LP is already highly learnable from structure and NC has headroom.
>
> ---
>
> ### Q2. Semi-supervised setting
>
> We did not study the extremely sparse-label regime in this paper, and do not want to overclaim. We chose 60/20/20 for NC specifically to study **cross-task transfer mechanisms rather than label-efficient learning**. With very sparse labels, it becomes difficult to disentangle genuine task incompatibility from simple data starvation, since NC may not learn sufficiently informative representations at all. The current model-selection guidelines should therefore be viewed as most reliable when NC has enough supervision to form task-specific geometry. Whether they remain predictive in few-shot settings is an important open question requiring a different experimental design.
>
> ---
>
> ### Q3. Automation for transfer framework
>
> Yes, this is a promising direction. Our paper takes a static first step by showing that **simple dataset diagnostics, especially homophily and baseline task learnability**, can already guide transfer direction and mechanism choice before training. Extending this to a dynamic framework that adjusts task coupling online is a natural next step, for example by combining our diagnostics with adaptive weighting methods such as GradNorm. Our diagnostics provide a natural signal for such adaptive weighting. We did not pursue this here because it introduces additional variance and design choices, whereas our goal was to first establish a clean protocol and reliable predictors.

---

> > ### Author Rebuttal · Reviewer_dY7S · 2026-04-01
> >
> > Thank you for the detailed response. I recognize the contribution of this work in establishing a standardized leakage-free evaluation protocol and will keep my original score.
> >
> > Meanwhile, I still have the following concerns:
> > When encountering heterophilic graph structures, should we simply abandon transfer learning entirely? I hope the paper can further explore active strategies to mitigate negative transfer, instead of only relying on simple statistical indicators for passive avoidance.
> > The generalization ability under sparse label scenarios still requires further validation.

---

> > > ### Author Response · Authors · 2026-04-02
> > >
> > > Thank you for the thoughtful follow-up and positive remarks. We agree that the goal is not to abandon transfer in heterophilic settings, but to apply it more carefully.
> > >
> > > Our results suggest that the issue is not transfer per se, but which mechanism is used. In heterophilic settings, we consistently observe that post hoc representation reuse, such as ET-Rep, is unstable and can lead to negative transfer, whereas *coupled approaches* such as MV and Joint are consistently more robust across datasets and backbones. We also emphasize that our paper does more than provide passive avoidance: MV and Joint are active coupled training strategies already evaluated in this work, and they provide a concrete positive recommendation: prefer coupled or alignment-based training that allows the model to reconcile task objectives during optimization, rather than directly reusing representations optimized for a different objective.
> > >
> > > More broadly, our findings indicate that negative transfer can arise from incompatibility between node-discriminative and edge-scoring objectives, which is amplified in heterophilic graphs. This points naturally toward stronger active mitigation strategies, such as controlled coupling, representation alignment, or adaptive weighting between tasks. Our protocol and diagnostics provide a clean foundation for evaluating such methods in future work.
> > >
> > > Regarding the sparse-label settings, we agree that this is an important direction. Our current conclusions are most reliable when NC has sufficient supervision to form meaningful representation geometry. In sparse-label settings, the challenge shifts toward *whether NC itself can learn informative features*, making transfer behavior harder to disentangle from data scarcity. We will clarify this limitation in revision and highlight sparse-label generalization as an important future direction.

---

### Official Review · Reviewer_4cta · 2026-03-11

**Soundness:** 3
**Presentation:** 4
**Significance:** 3
**Originality:** 3
**Overall Recommendation:** 4
**Confidence:** 3

**Summary:**

This paper studies same-graph cross-task transfer between node classification (NC) and link prediction (LP) in graph neural networks. The authors identify a core problem: existing evaluations of NC and LP use incompatible splits, observed graphs, and negative sampling strategies, making it impossible to draw reliable conclusions about cross-task transfer. To address this, they propose a leakage-free evaluation protocol that fixes node/edge splits, uses a shared message-passing graph excluding evaluated edges, and employs fixed negatives for LP. Under this protocol, they evaluate five transfer mechanisms (Warm Start, ET-Rep, ET-Concat, Multi-View, Joint) across three backbones (GCN, GraphSAGE, GPS) on 11 datasets spanning homophilic, heterophilic, and mixed regimes. They find that NC→LP transfer is reliably beneficial on homophilic graphs, while LP→NC is fragile and regime-dependent. They also introduce the CoTask Score (CTS) to summarize joint NC+LP utility, and show that simple statistics like homophily can predict when and in which direction transfer helps.

**Compliance With Llm Reviewing Policy:**

Affirmed.

**Final Justification:**

I'll keep my scores.

**Key Questions For Authors:**

1. Many reported gains in Tables 2–3 are on the order of 0.1–2 percentage points. Could the authors provide standard deviations across random seeds for the main results, and clarify which gains are statistically significant? I think it's important for assessing whether the protocol differences actually matter at the reported scale.
2. The paper identifies USA/Europe/Brazil as "structure-dominant" based on a qualitative description (high LP learnability, NC headroom), but does not give a precise criterion. This is kinda not rigorous. Could the authors provide a formal definition or quantitative threshold (e.g., based on baseline LP AUC and NC accuracy under a fixed backbone) that practitioners could apply to new datasets? How would this regime interact with larger, feature-rich graphs?
3. The alignment weight $\lambda_x$ in MV is set via a formula (Eq. 1) that uses graph statistics directly rather than validation tuning. Does this formula favor homophilic graphs by construction? May I ask if you have compared this heuristic schedule against standard validation-based grid search for $\lambda_x$, and is MV's reported performance sensitive to this choice?

**Limitations:**

Partially, they do have a impact section discussing potential positive/negative impact, but maybe they should also acknowledge: (1) the small number of datasets and their limited size diversity; (2) the incomplete inductive evaluation; (3) the restriction to NC and LP only, excluding graph-level tasks; and (4) the absence of statistical significance testing for gain tables. Adding a short limitations paragraph to the conclusion would make the paper more complete. Conclusively, the impact statement covers societal concerns but not methodological scope limitations.

**Strengths And Weaknesses:**

Strengths:

1. Soundness. The paper addresses a genuine method gap. The leakage-free protocol is well-motivated and technically careful: fixing LP negatives, excluding evaluation edges from message passing, and using shared splits are all necessary conditions for fair cross-task comparison. The sanity checks in Appendix A.2 (verifying no edge leakage, no overlap between negatives and positives, reproducibility) add confidence in the implementation. The finding that NC→LP transfer correlates strongly with homophily (Pearson r up to 0.904 for ET-Concat/GCN on node homophily) is well-supported statistically and consistent across all three backbones, which kinda support the directional asymmetry claim. The distinction between homophilic, heterophilic, and structure-dominant setup is a useful organizing principle.
2. Presentation. The paper is clearly written and well-structured. The problem motivation is crisp, the protocol is stated precisely with formal notation, and the transfer mechanisms are described in a principled way using the shared parameter-space framing (Figure 1). The connection between mechanism type (parameter reuse vs. representation reuse vs. coupled training) and expected behavior is intuitive and helps readers understand why results differ across regimes. Tables are well-organized and the narrative in Section 5 does a good job connecting results back to the paper's claims.
3. Significance. The problem is practically important. Many deployed graph systems serve both NC and LP objectives over the same graph, and practitioners routinely face the question of whether to train a shared encoder. The paper's finding that ET-Rep (frozen representation reuse) is dangerous while MV/Joint are more robust is immediately actionable. The CoTask Score provides a useful reporting primitive that future benchmarking papers could adopt. The protocol itself is a contribution independent of the empirical findings, as it gives future researchers a clean way to study same-graph multi-task learning.
4. Originality. The specific combination of same-graph, bidirectional NC↔LP transfer, unified leakage-free protocol, regime-based predictors is novel. Prior work on multi-task graph learning typically proposes new architectures or pretraining objectives evaluated on one downstream task. The paper's contribution is explicitly not architectural; it is evaluative and diagnostic, and their observations are original.

Weaknesses:
1. Soundness. First, the study covers only 11 datasets, of which 3 are homophilic (Cora, Citeseer, Pubmed), which are a narrow and well-studied slice. The homophilic finding (NC→LP works) rests heavily on these three citation graphs, which share similar structure (sparse, moderate size, bag-of-words features). It is unclear how robust the correlation with homophily would be across a broader and more diverse set of graphs with varying sizes, densities, and feature types. The significance tests in Table 5 are over n=11 datasets, a very small sample for Pearson correlation. Second, the paper excludes MV and Joint from the inductive setting results, citing future work. This is a meaningful gap: MV and Joint are the most consistently recommended mechanisms in the transductive analysis, but their inductive behavior is unknown. This weakens the practical guidance for deployment settings where inductive generalization matters. The structure-dominant regime (USA, Europe, Brazil) is identified post-hoc as the setting where LP→NC works. The definition of this regime, ie, "high LP learnability with substantial NC headroom", is not operationalized with a precise threshold or criterion, making it hard for practitioners to reliably identify this regime on new graphs. Third, I feel like the choice of $\lambda_x$ for the MV model (Equation 1 in the appendix) is set using a formula involving dataset statistics rather than validation tuning. While the authors argue this removes a trained parameter, it introduces a strong inductive bias that implicitly assumes homophilic graphs should always be coupled more aggressively. This could confound MV results, i.e., it is unclear whether MV's performance reflects the alignment objective or this graph-statistic-based hyperparameter schedule.

2. Presentation. First, the paper does not include variance (standard deviation or confidence intervals) across random seeds in the main tables. Given that results are reported as gains in percentage points (sometimes as small as 0.1–0.4 pp), it is difficult to assess whether reported improvements are statistically meaningful or within noise. Second, the "structure-dominant" regime is described narratively but never formally defined in the main text, making it a bit hard for me to follow. Third, I think the Figure 1 can be improved, at least improve its resolution and make it a vector figure.

3. Significance. First, the paper focuses exclusively on the transductive setting in its main results, with inductive results pushed to an appendix that is incomplete (missing MV and Joint). Many real-world graph s require inductive generalization (e.g., new users joining a social network), so this limits the scope of the guidance. Second, the datasets used for the mixed/structure-dominant regime (USA, Europe, Brazil airport graphs) are quite small (131–1190 nodes) and lack node features, which may limit generalizability to richer real-world graphs.

4. Originality. I think the work has good originality, as I have outlined in the Strengths part. I couldn't clearly identify its weakness in oriignality.

---

> ### Author Rebuttal · Authors · 2026-03-28
>
> **Thank you for the thoughtful review. We address the main concerns below.**
>
> ### W1a. Diversity of homophilic datasets
>
> We agree broader homophilic coverage would strengthen the empirical picture. Our conclusion, however, does not rest only on three citation graphs. It rests on **a repeated directional pattern across all 11 benchmarks**, multiple backbones, and mechanisms, with Pearson correlation as supporting evidence. To directly address the diversity concern, we added **Amazon-Photo** and **Amazon-Computers** (dense co-purchase graphs, He=0.83/0.78, Hn=0.85/0.80). The pattern holds on both: NC→LP gains are positive across all mechanisms while LP→NC is weak and inconsistent, with ET-Rep showing substantial negative transfer (-4.3 pp on Photo, -14.3 pp on Computers), further confirming that **the finding is not specific to sparse citation graphs**.
>
> | Dataset | Task | Base | WS | ET Rep | ET Con | MV | Joint |
> |---|---|---|---|---|---|---|---|
> | Photo | NC→LP (AUC) | 94.6 ± 1.0 | +2.5 ± 0.8 | **+3.2 ± 0.1** | +2.7 ± 0.3 | +1.6 ± 0.2 | +2.8 ± 0.1 |
> | Photo | LP→NC (Acc) | 93.0 ± 1.6 | +0.8 ± 0.7 | -4.3 ± 0.9 | +0.6 ± 0.7 | -0.1 ± 0.8 | **+1.0 ± 0.7** |
> | Computers | NC→LP (AUC) | 93.7 ± 0.8 | +0.2 ± 1.0 | **+3.8 ± 0.1** | +2.2 ± 0.3 | +1.4 ± 0.6 | +2.5 ± 0.5 |
> | Computers | LP→NC (Acc) | 89.1 ± 0.7 | **+0.0 ± 0.3** | -14.3 ± 2.3 | -0.6 ± 1.1 | -1.3 ± 0.6 | -0.2 ± 1.1 |
>
> ---
>
> ### W1b / W3a. Inductive setting and missing MV/Joint
>
> We agree this was an important gap. We have now run MV and Joint in the inductive setting for GCN, and **the same qualitative conclusion holds**: NC→LP remains strongly positive on homophilic datasets, while LP→NC remains much less reliable. For NC→LP, MV/Joint gains are 20.31/21.37 on Cora, 22.26/19.79 on Citeseer, and 16.51/16.34 on Pubmed. In contrast, **LP→NC remains small and inconsistent** on the same datasets: 1.56/-0.40 on Cora, 1.90/-0.92 on Citeseer, and -0.47/-0.11 on Pubmed. Due to the rebuttal space limit, we cannot include the full table here, but we will provide the complete inductive MV/Joint results in the camera ready version.
>
> ---
>
> ### W1c / Q3. Alignment weight λx in MV
>
> We agree that the appendix wording can be misleading. In the reported experiments, **λx is selected by validation tuning**, as stated in Section 4.3. Eq. 1 in Appendix A.1 is used only to initialize the search range, not to determine the final reported λx. This substantially reduces the confounding concern: the homophily-based formula biases the search toward a reasonable region of [0.2,1.6], but the final value is selected by validation, with no access to test performance. Moreover, the same asymmetry appears for WS, ET-Rep, ET-Concat, and Joint, **none of which use homophily-based hyperparameter initialization**, so the pattern is unlikely to be an MV-specific artifact.
>
> ---
>
> ### W2a / Q1. Standard deviations and significance
>
> Standard deviations across seeds were computed for all runs and omitted from the main tables only for readability; we will include them in the appendix of the revision. As already reflected in the Amazon-Photo and Amazon-Computers results and sensitivity tables in this rebuttal, variability is well-behaved and the directional pattern is stable across seeds, with the larger gains on homophilic datasets consistently exceeding the run-to-run variation. Our conclusions do not rely on marginal 0.1–0.4 pp changes, which occur mostly in regimes where transfer is not expected to help and should be interpreted cautiously. We will also add significance tests in the revision.
>
> ---
>
> ### W2b / Q2. Structure-dominant regime
>
> "Structure-dominant" is a regime descriptor for datasets where **structure strongly predicts edge formation while NC remains unsaturated**. Under a fixed backbone and protocol, a concrete operationalization is: baseline LP AUC ranks in the upper half of the benchmark while baseline NC accuracy ranks in the lower half. This avoids an arbitrary absolute threshold and makes the notion reproducible. For larger, feature-rich graphs, the same concept can still apply when features are abundant yet weakly aligned with labels while graph structure remains highly informative for edge formation.
>
> ---
>
> ### W2c. Figure quality
>
> Thanks for the comment. We will replace Figure 1 with a clearer vector figure if accepted.
>
> ---
>
> ### W3b. Small mixed-regime graphs
>
> We agree that USA, Europe, and Brazil are relatively small and feature-poor. They are included not because airport graphs represent all real-world deployments, but because **they cleanly instantiate the regime** relevant to our claim: LP is highly learnable from structure while NC remains unsaturated. This makes them useful stress tests for the LP→NC structural pretraining effect we document. We acknowledge that larger, feature-rich datasets with the same joint profile would strengthen generality.

---

> > ### Author Rebuttal · Reviewer_4cta · 2026-04-01
> >
> > My concerns have been adequately addressed. As I already have positive score, I would like to maintain it.

---

> > > ### Author Response · Authors · 2026-04-02
> > >
> > > Thank you for the insightful and constructive review, and for the positive assessment of our work. We truly appreciate your detailed feedback and thoughtful suggestions, which helped us improve the clarity, rigor, and overall presentation of the paper.

---

### Official Review · Reviewer_fBt6 · 2026-03-12

**Soundness:** 3
**Presentation:** 2
**Significance:** 2
**Originality:** 2
**Overall Recommendation:** 4
**Confidence:** 3

**Summary:**

This paper investigates the mechanism of cross-task transfer between node classification (NC) and link prediction (LP) using graph neural networks (GNNs) on the same graph. It points out that conclusions drawn in prior evaluations are often unreliable due to data leakage and inconsistent data splits, and accordingly proposes a strict leakage-free evaluation protocol. Based on this more rigorous benchmark, the paper reveals a strong directional asymmetry in cross-task transfer: on homophilous graphs, transferring knowledge from NC to LP typically yields consistent gains, whereas transfer from LP to NC is much less robust and becomes effective only under a “structure-dominated” regime, where the LP task is relatively easy and the NC task still has room for improvement, allowing LP to serve as a form of graph-structure pretraining. In addition, the paper introduces a dimensionless composite metric, the CoTask Score (CTS), and provides useful empirical guidance and a more reliable evaluation paradigm for the deployment, assessment, and selection of shared encoders in single-graph multi-task learning.

**Compliance With Llm Reviewing Policy:**

Affirmed.

**Final Justification:**

The paper addresses a key weakness in current multi-task evaluation for graph neural networks (GNNs), namely spurious gains caused by protocol artifacts and data leakage. The proposed leakage-free protocol—by fixing node/edge splits, strictly excluding evaluation edges, and standardizing the negative sampling procedure—effectively eliminates the systematic biases that have affected prior cross-task evaluation on the same graph. The experimental design across multiple backbones, including GCN, GraphSAGE, and GPS, is very solid.

Initially, I raised concerns about cross-graph settings, and experiments with limited backbones. The authors argued that these are beyond the scope of the main problems, and it would be difficult to conduct these exploration in a limited rebuttal window.

I encourage the authors to include some of their rebuttal (e.g., regarding backbones) into future work discussions.

**Key Questions For Authors:**

1, The current conclusions are established only in the same-graph setting. Since an important direction of modern graph foundation models is zero-shot generalization to unseen graphs, it would significantly strengthen the paper if the authors could discuss whether the proposed predictors and the observed NC/LP asymmetry still hold under cross-graph distribution shift, such as large changes in homophily. Preliminary evidence of the likely generalization boundary would make the work more significant and forward-looking, and would positively affect my rating.

2. The current study focuses on relatively standard backbones such as GCN, SAGE, and GPS. It would strengthen the paper if the authors could discuss whether the same transfer asymmetry is expected to persist under newer architectures with stronger long-range modeling ability, such as sequence-based or state-space-inspired graph models (Such as Graph Transformer). Even a small-scale ablation would improve the paper’s frontier relevance and make me more positive.

3. A strength of the paper is that it predicts when negative transfer is likely to occur, but it would be even more impactful if the authors could go one step further and discuss possible intervention strategies, such as feature disentanglement or contrastive alignment, to reduce the incompatibility between NC and LP representations. Showing even a simple mitigation strategy that improves a difficult LP →\rightarrow→ NC case would substantially increase the methodological value of the work.

**Limitations:**

Yes

**Strengths And Weaknesses:**

S1. The paper addresses a key weakness in current multi-task evaluation for graph neural networks (GNNs), namely spurious gains caused by protocol artifacts and data leakage. The proposed leakage-free protocol—by fixing node/edge splits, strictly excluding evaluation edges, and standardizing the negative sampling procedure—effectively eliminates the systematic biases that have affected prior cross-task evaluation on the same graph. The experimental design across multiple backbones, including GCN, GraphSAGE, and GPS, is very solid.

S2. Although the paper does not propose a new neural architecture, it offers valuable new insights through a deep evaluation of existing methods. In particular, the paper provides a careful analysis of the directional asymmetry between node classification (NC) and link prediction (LP): transfer from NC to LP is typically beneficial, whereas transfer from LP to NC is much more fragile and can easily lead to negative transfer. More importantly, it introduces and characterizes a structure-dominant regime, showing that when LP is relatively easy while NC still has room for improvement, LP can serve as a effective form of structural pretraining.

S3. This work has strong practical relevance for both industry and the graph representation learning community. The paper introduces the CoTask Score (CTS), a dimensionless composite metric that provides a standard tool for evaluating the joint performance of shared encoders across multiple tasks. In addition, the paper shows that the success or failure of transfer is at least partly predictable: practitioners can use simple graph statistics such as homophily to anticipate and avoid negative transfer in advance.

S4. The paper is clearly structured and easy to follow. The overall narrative develops naturally from identifying flaws in prior evaluation protocols, to establishing a principled benchmark, to conducting empirical analysis, and finally to proposing predictive indicators. The discussion of related work is also clear.

W1. The paper restricts its scope to the same-graph setting. However, a major direction in current large graph models is moving away from performance optimization on a single graph and toward stronger cross-graph and cross-domain zero-shot generalization. It therefore remains unclear whether the paper’s conclusions can extend smoothly to multi-task settings under cross-graph distribution shift or out-of-distribution (OOD) conditions.

W2. The evaluated backbones (GCN, GraphSAGE, and GPS) are relatively standard. Meanwhile, the field is seeing a growing number of newer architectures with stronger long-range dependency modeling capabilities, including recent sequence-inspired graph models. These architectures rely on substantially different mechanisms for capturing topology, so it is unclear whether the paper’s claimed asymmetry pattern in cross-task transfer, especially between LP and NC, would still hold for them.

W3. When explaining negative transfer or directional asymmetry, the paper mainly relies on intuitive arguments based on the optimization landscape in parameter space. It lacks a more rigorous theoretical treatment from perspectives such as graph signal processing, generalization error analysis, or feature-space alignment.

W4. The paper’s main takeaway is that negative transfer can be predicted from simple graph statistics, but it does not go further to propose concrete algorithmic interventions. For example, it does not explore whether techniques such as feature disentanglement or contrastive alignment could be used to actively reduce the incompatibility between NC and LP in representation space. This somewhat limits its methodological impact as a top-conference contribution.

---

> ### Author Rebuttal · Authors · 2026-03-28
>
> **We thank the reviewer for their valuable and insightful feedback. We address the main concerns below.**
>
> ### W1/Q1. Same-graph scope and cross-graph generalization
>
> We appreciate this perspective, but view it as a question of scope rather than a weakness. The same-graph setting is not an overlooked limitation. It is the precise and intentional scope of our contribution, addressing a well-defined and practically important problem: when NC and LP coexist on a single graph, should one reuse representations across tasks, and in which direction? Same-graph transfer and cross-graph generalization are **complementary directions, not competing ones**. Before asking whether transfer patterns persist under distribution shift across graphs, one must first establish what those patterns are on a fixed graph under controlled conditions. Our paper provides exactly that foundation. We also note that much of the recent cross-graph and foundation-model literature focuses on transferring a single downstream task across graphs, rather than the same-graph multi-task interaction studied here. Our setting is therefore **orthogonal and currently underexplored**. We agree that extending the framework to cross-graph or OOD settings is an important future direction and will discuss this explicitly in the revision.
>
> ---
>
> ### W2/Q2. Newer backbone architectures
>
> We chose GCN, GraphSAGE, and GPS deliberately to span the main architectural families: GCN as a canonical message-passing baseline, GraphSAGE as an inductive neighborhood-aggregation model, and GPS as a graph transformer with global attention. The fact that **the NC→LP asymmetry appears consistently across all three**, despite their different mechanisms for capturing topology, suggests that the pattern is not tied to a single backbone family. We also note that many newer graph architectures still build on these same core ingredients, typically by extending message passing, attention, or their combination, rather than changing the underlying tension between node-discriminative and edge-scoring objectives. We agree that sequence-inspired or state-space graph models are an important frontier, and we do not claim universality beyond the architectures studied here. Our point is narrower: the asymmetry is already robust across representative message-passing and transformer-style backbones. We will clarify this scope in the revision.
>
> ---
>
> ### W3. Rigorous analysis
>
> We respectfully disagree that the paper lacks rigor because it does not include a formal theory. The contribution is to formalize the same-graph NC↔LP transfer problem, remove protocol confounds that invalidate prior comparisons, and **establish a robust directional pattern under controlled evaluation**. This is exactly the kind of question that must be settled empirically before a clean theory can be meaningfully built. The explanation is also not merely intuitive. The paper provides two complementary mechanism-level views: a shared latent-space view, where NC and LP differ in how they read out the same encoder geometry, and a shared parameter-space view, where the two tasks induce different but potentially compatible or conflicting losses over the same encoder family. These views directly explain the observed pattern. That is already **a rigorous, mechanism-backed empirical result**. A graph signal processing analysis or generalization bound would be interesting future work, but the absence of one formalization does not make the current contribution insufficiently rigorous.
>
> ---
>
> ### W4/Q3. Algorithmic interventions
>
> Thank you for this important comment. We view this as complementary rather than required for this paper. Establishing a clean evaluation paradigm and extracting actionable predictors is a deliberate contribution, not a limitation. There was no standardized leakage-free protocol for same-graph NC↔LP transfer, and that foundation is a prerequisite for the algorithmic work suggested. We also explored additional transfer mechanisms beyond the five reported, including *prompt-based transfer* and *gated fusion variants*. In our preliminary experiments, these did not show consistent improvements over the simpler mechanisms and were dropped to keep the study focused and interpretable. This experience actually reinforces one of the paper's main messages: **the difficulty of LP→NC transfer is not easily resolved by adding architectural complexity**, and the regime-aware guidance we provide is therefore more actionable than a new mechanism that works only in specific cases. We consider feature disentanglement and contrastive alignment natural and important directions for future work, with our protocol and diagnostics providing the foundation on which such methods can be cleanly evaluated.

---

> > ### Author Rebuttal · Reviewer_fBt6 · 2026-04-03
> >
> > The rebuttal helps clarify the intended scope and strengthens the empirical motivation, but it does not fully resolve my main concerns. In particular, the paper still lacks direct evidence on cross-graph/OOD generalization, additional validation on newer backbone families, and any concrete intervention beyond predictive diagnostics, which were the main issues behind my original assessment.
> >
> > Overall, I appreciate the authors’ response, but I do not think it changes my evaluation enough. I therefore maintain my original score.

---

> > > ### Author Response · Authors · 2026-04-03
> > >
> > > Thank you for the thoughtful follow-up. We would like to address the three remaining concerns directly, since we believe they concern either the scope of the paper or contributions already present in the current work.
> > >
> > > ### On cross-graph/OOD generalization
> > >
> > > We respectfully maintain that same-graph transfer and cross-graph generalization address **distinct research questions**, rather than the same question at different scales. **Studying same-graph transfer under cross-graph distribution shift** would introduce an additional source of variation and make it difficult to isolate the transfer mechanism itself, which is precisely what our protocol is designed to control. We note that foundational single-graph work, including Kipf & Welling (GCN), Hamilton et al. (GraphSAGE), and Zhang & Chen (SEAL), established results on fixed graphs before later extensions to broader transfer settings, and those contributions are not considered incomplete for that reason. Our paper is intended in the same spirit for same-graph NC↔LP transfer: it establishes a clean and controlled foundation on top of which cross-graph and OOD extensions can be studied rigorously. Cross-graph generalization is an important future direction that this paper's controlled foundation makes tractable, but it is not part of the core problem setting of the present work.
> > >
> > > ### On newer backbone architectures
> > >
> > > Our goal was to test whether the observed asymmetry is tied to a specific model family. GCN, GraphSAGE, and GPS were chosen deliberately to span representative message-passing and transformer-style architectures, and the same directional pattern appears consistently across all three despite their substantially different mechanisms for capturing topology. GPS in particular already includes global attention and is architecturally closer to newer long-range models than standard message-passing baselines. We agree that broader backbone coverage would strengthen the empirical picture. At the same time, a meaningful study of additional architectures would require **a full sweep across 11 datasets, 5 transfer mechanisms, and both transfer directions, which was not feasible within the rebuttal window.** We therefore prioritized the additional experiments that addressed the most central empirical gaps raised across reviewers, and would plan to expand backbone coverage in the camera-ready version if accepted.
> > >
> > > ### On concrete interventions
> > >
> > > We would also like to clarify that the paper does not only provide predictive diagnostics; **it already proposes and evaluates concrete intervention strategies.** MV and Joint are active coupled training mechanisms, not diagnostic tools. MV uses InfoNCE-based cross-view alignment between NC and LP encoders, which already instantiates the kind of contrastive alignment strategy the reviewer suggests. Joint training couples NC and LP objectives through a shared encoder and adaptive loss weighting. Across datasets and backbones, **these coupled strategies consistently reduce the risk of negative transfer relative to naive post hoc reuse**, yielding a clear practical recommendation: when NC and LP objectives may be incompatible, prefer coupled or alignment-based training over direct representation reuse. The paper thus contributes both diagnostic guidance about when transfer is likely to help and algorithmic guidance about which transfer mechanisms are more reliable; these are complementary contributions, not substitutes for each other.
> > >
> > > We appreciate that the reviewer recognized the rigor of the protocol and the practical relevance of our findings. Overall, we view this paper as making **three concrete contributions** within its intended scope: **a leakage-free measurement framework for same-graph NC↔LP transfer**, a systematic empirical characterization of the **directional asymmetry across datasets and backbones**, and **practical intervention guidance** showing that coupled strategies such as MV and Joint are more reliable than naive reuse when negative transfer is a risk. The remaining concerns mainly point to important extensions beyond the present setting, rather than missing evidence for the central claims we make here. We hope this clarifies that the paper does not merely diagnose an existing setting, but helps **establish same-graph NC↔LP transfer as a well-posed problem in the literature**, together with a methodological framework and practically useful guidance for studying it.

---

### Official Review · Reviewer_RiUD · 2026-03-13

**Soundness:** 2
**Presentation:** 2
**Significance:** 3
**Originality:** 3
**Overall Recommendation:** 4
**Confidence:** 4

**Summary:**

This paper formalizes same-graph cross-task transfer between node classification and link prediction in GNNs and proposes a leakage-free evaluation protocol. The key empirical finding is that transfer is strongly directional: NC→LP is reliably beneficial on homophilic graphs, while LP→NC is fragile and only effective in a "structure-dominant" regime where LP is easy but NC has headroom. A CoTask Score metric is also introduced to summarize joint NC+LP utility under a shared encoder.

**Compliance With Llm Reviewing Policy:**

Affirmed.

**Final Justification:**

The author addressed my concerns and I think the paper presents an intriguing research direction. I decide to increase my score and hope the author to integrate the important points in the paper.

**Key Questions For Authors:**

1. Can you clarify the quantitative criteria for a dataset to be classified as "structure-dominant"? Specifically, what thresholds on LP learnability and NC headroom define this regime?

2. What is the training time overhead of MV and Joint relative to single-task baselines?

3. Can you provide any formal analysis, even under a simplified model, explaining why NC supervision produces embeddings compatible with LP on homophilic graphs? The current explanation is entirely intuitive.

**Limitations:**

yes

**Strengths And Weaknesses:**

### Strengths:

This paper reveals a directional asymmetry in cross-task transfer on graphs, which offers valuable insights for current research on pretraining graph models. The authors also propose a reasonable evaluation protocol for cross-task transfer. Fixing splits, negatives, and excluding evaluation edges from message passing addresses real confounding issues in existing cross-task evaluations. Overall, this paper provides an interesting perspective as well as an empirical benchmark for the design of cross-task graph models.

### Weaknesses

However, this paper remains somewhat surface-level, serving more as a diagnostic study. It does not deeply investigate why cross-task transfer is related to homophily, or why embeddings from different tasks are compatible on homophilic graphs.

1. The paper claims that cross-task transfer is directional and predictable, but this is validated using only two tasks. It remains unclear whether such directional asymmetry also exists across other graph tasks (e.g., graph classification, anomaly detection, or community detection).

2. The paper extensively discusses unique findings about cross-task transfer on homophilic graphs, yet never provides a formal definition of homophily throughout the manuscript, making the presentation incomplete.

3. It is unclear whether the proposed methods are sensitive to hyperparameters. The authors should provide more experimental results to demonstrate the stability of the approach, such as hyperparameter sensitivity analysis and standard deviations across runs.

---

> ### Author Rebuttal · Authors · 2026-03-28
>
> **We thank the reviewer for their valuable and insightful feedback. We address the main concerns below.**
>
> ### W0. Diagnostic study / mechanism depth.
>
> We view this differently. Diagnostic and protocol-setting work is especially important when a phenomenon is easy to mismeasure, which is exactly the case here. Our contribution is not just to report a trend, but to formalize a previously uncontrolled transfer setting, remove confounds through a **leakage-free benchmark**, show that the asymmetry is genuine rather than a protocol artifact, and connect it to actionable predictors through CTS and regime-aware mechanism selection. The paper also goes beyond description: Section 4 and Appendix A.4 explain why NC→LP is favored on homophilic graphs. In particular, the shared latent-space and shared parameter-space views explain **how NC supervision organizes embeddings into label-consistent local geometry useful for edge scoring**, whereas LP supervision does not necessarily induce class-discriminative structure, making the reverse direction less reliable. A formal theorem would be valuable future work, but the present contribution is already more than descriptive.
>
> ---
>
> ### W1. Other task families.
>
> We respectfully think this asks the paper to cover a broader scope than intended. **Our claim is not that all task pairs exhibit the same asymmetry**; it is that NC and LP form the most natural and practically important same-graph pair, with supervision signals that naturally coexist on many labeled graphs, and this case had not been studied under a clean leakage-free protocol. Some examples listed are not direct extensions: graph classification is defined over collections of graphs, not multiple objectives on one fixed graph. **Community detection and subgraph prediction raise harder protocol questions** around shared observed structure, leakage across overlapping subgraphs, and compatible splits, and these require a separate problem formulation. We will clarify that our claims are specific to NC↔LP and position other task families as future work.
>
> ---
>
> ### W2. Homophily definition.
>
> Thank you for sharing this point. Edge homophily: $H_e = |\{ (u,v) \in E : y_u = y_v \} | / |E|$. Node homophily: $H_n = \frac{1}{|V|} [\sum_{v} | \{ u \in \mathcal{N}(v) : y_u = y_v \} | / |\mathcal{N}(v)|]$. Both are already reported in Table 1; we will add formal definitions and a brief intuitive explanation to the main text and appendix.
>
> ---
>
> ### W3. Sensitivity / standard deviations.
>
> Due to space limitations, **please refer to our response to Reviewer 4cta - W2a**.
>
> ---
>
> ### Q1. Structure-dominant regime.
>
> "Structure-dominant" is a regime description, not a hard-threshold taxonomy. It refers to datasets where **structure strongly predicts edge formation while NC remains unsaturated**; empirically, high baseline LP AUC with substantial NC headroom. We avoided a universal numeric cutoff because LP AUC and NC accuracy are not comparable across datasets and backbones. In the revision we will operationalize it via **relative baseline rankings under a fixed protocol**: datasets where LP AUC ranks in the upper half and NC accuracy in the lower half across our benchmark.
>
> ---
>
> ### Q2. Training time overhead.
>
> For Cora, GCN backbone, averaged over 10 seeds on a local workstation (Apple M3 Pro, 12-core CPU, 18-core GPU, 18 GB RAM):
>
> | Method | Time (s) |
> |--------|----------|
> | NC base | 0.69 ± 0.05 |
> | LP base | 2.29 ± 0.51 |
> | MV | 5.25 ± 1.76 |
> | Joint | 19.76 ± 2.72 |
>
> **Joint's overhead is dominated by the λ grid search** (11 values); per-run cost is ~1.80s, comparable to MV. We will report this breakdown and additional datasets in the revision.
>
> ---
>
> ### Q3. Formal analysis for NC→LP.
>
> We acknowledge the current explanation is mechanistic rather than a formal theorem and will make that explicit in the revision. To go beyond intuition, **we ran a representational geometry analysis**: for each dataset, we computed the mean cosine similarity gap between linked and unlinked pairs using raw features vs. NC-learned embeddings (GCN, ET-Rep). As shown in the table below, NC training amplifies the alignment gap across all datasets, but **the resulting embedding gap is ~2.7x larger on homophilic graphs** (mean 0.79) than heterophilic ones (mean 0.29). This provides quantitative evidence for why NC→LP transfer gains are larger on homophilic graphs: NC supervision shapes a geometry where linked pairs are far more separable, making learned embeddings naturally compatible with LP decoding. We will include this analysis in detail in the revision.
>
> | Dataset | Hn | Feat Gap | Emb Gap | Increase |
> |---|---|---|---|---|
> | Texas | 0.057 | 0.0155 | 0.2681 | +0.2526 |
> | Cornell | 0.111 | 0.0272 | 0.3218 | +0.2946 |
> | Wisconsin | 0.155 | 0.0331 | 0.2793 | +0.2462 |
> | Actor | 0.220 | -0.0081 | 0.2920 | +0.3001 |
> | CiteSeer | 0.706 | 0.1475 | 0.8932 | +0.7457 |
> | PubMed | 0.792 | 0.1991 | 0.6902 | +0.4911 |
> | Cora | 0.825 | 0.1120 | 0.7876 | +0.6756 |

---

> > ### Author Rebuttal · Reviewer_RiUD · 2026-04-06
> >
> > I appreciate the detailed clarifications for all my questions. I have increased my score to 4 and I recommend the authors to include them in their final draft to make it for the reader to better understand the motivation and method.

---

> > > ### Author Response · Authors · 2026-04-07
> > >
> > > Thank you for the follow-up and for increasing your score. We are glad the clarifications were helpful, and we will incorporate them more clearly in the final draft.

---

### Decision · Program_Chairs · 2026-04-30

**Decision:**

Accept (regular)

**Comment:**

The paper investigates the same graph cross-task transfer between node classification and link prediction. The authors propose a rigorous, leakage-free evaluation protocol to address inconsistencies in prior evaluations. By evaluating multiple transfer mechanisms, the authors reveal a strong directional asymmetry: transfer from node classification to link prediction is consistently beneficial on homophilic graphs, whereas the reverse is fragile. The introduction of the CoTask Score and the use of homophily as a predictor provide practical value.

The reviewers unanimously agreed on the methodological rigor and the practical relevance of establishing a standardized benchmark. During the rebuttal, the authors successfully addressed key concerns by providing additional evaluations on diverse datasets, including standard deviations, and presenting inductive setting results. Given the solid empirical foundation and actionable insights for the community, the paper is strongly recommended for acceptance.